# Strong Equilibria in Bayesian Games with Bounded Group Size

## Abstract

We study the group strategic behaviors in Bayesian games. Equilibria in previous work do not consider group strategic behaviors with bounded sizes and are too "strong" to exist in many scenarios. We propose the ex-ante Bayesian $k$-strong equilibrium and the Bayesian $k$-strong equilibrium, where no group of at most $k$ agents can benefit from deviation. The two solution concepts differ in how agents calculate their utilities when contemplating whether a deviation is beneficial. Intuitively, agents are more risk-averse in the Bayesian $k$-strong equilibrium than in the ex-ante Bayesian $k$-strong equilibrium. With our solution concepts, we study collusion in the peer prediction mechanisms, as a representative of the Bayesian games with group strategic behaviors. We characterize the thresholds of the group size $k$ so that truthful reporting in the peer prediction mechanism is an equilibrium for each solution concept, respectively. Our solution concepts can serve as criteria to evaluate the robustness of a peer prediction mechanism against collusion. Besides the peer prediction problem, we also discuss two other potential applications of our new solution concepts, voting and Blotto games, where introducing bounded group sizes provides more fine-grained insights into the behavior of strategic agents.

## CCS Concepts

• **Theory of computation → Solution concepts in game theory**; **Algorithmic mechanism design**; **Algorithmic game theory**.

## Keywords

Algorithmic Game Theory, Peer Prediction

**ACM Reference Format:**
Anonymous Author(s). 2025. Strong Equilibria in Bayesian Games with Bounded Group Size. In *Proceedings of The Web Conference 2025 (Web-Conf '25)*. ACM, New York, NY, USA, 23 pages. https://doi.org/XXXXXXX.XXXXXXX

## 1 Introduction

The Bayesian game model [30] is a powerful theoretical tool for analyzing agents' strategic behavior with incomplete information. It has been applied to a wide range of real-world scenarios, including auctions, eliciting information, marketing, and collective decision-making, just to name a few. In a Bayesian game, rational agents receive private information (named *types*) and act strategically to optimize the outcome in expectations conditioned on their types.

In Bayesian games, rational and self-interested agents may behave strategically and deviate from the intentions of the mechanisms. Moreover, in many real-world scenarios, agents may coordinate their strategic behavior to collectively benefit. For example, bidders may coordinate to place low bids in the auction and drive down prices, voters may collaborate to cast votes strategically, and agents may conspire to gain a higher payment in peer prediction systems.

**Example 1 (Group strategic behavior in peer prediction).** *Consider an online crowdsourcing (for example, image labeling) group. A peer prediction mechanism is applied to evaluate the quality of worker reports and calculate their rewards. The reward of a worker is calculated by comparing his/her report with another worker's (called a peer) report.*

*In a peer prediction mechanism, a worker usually gets a higher reward when his/her report has higher agreement with the peer's report. Therefore, a group of workers can benefit by colluding in advance and reporting the same answer to receive higher payments. However, the information collector desires to design a mechanism to prevent collusion and collect truthful reports from agents.*

Therefore, the importance lies in answering the following research question:

### How can we predict the outcome of Bayesian games with group strategic behavior?

Previous literature [26, 33, 53] have developed "strong" or "coalitional" equilibria, in an analogy of strong Nash equilibrium [5], to predict group strategic behaviors in the Bayesian game model, in which no group of agents shall benefit from strategic deviation. However, most solution concepts allow an arbitrary size of the strategic group and fall into the same criticism of "being too strong" as the strong Nash equilibrium. For example, Gao et al. [24] show that truth-telling cannot be a strong equilibrium in many peer prediction mechanisms, and Han et al. [28] show that a strong equilibrium may not exist in majority voting with incomplete information.

On the other hand, group strategic behaviors usually happen with a bounded size in real life. For example, a player will only collude with his/her friends or trusted players, and a mechanism that prevents deviations from a bounded size of groups is sufficient to collect truthful reports in most cases [57]. Moreover, instead of the "all-or-nothing" characterizations, studying equilibria with coalitions of a bounded size provides richer structures that enable more constructive solutions to many economic problems. However, whether a bounded size of group deviation exists is not characterized by the "strong" equilibria in the previous work.

### 1.1 Our Contribution

We propose two solution concepts in which group strategic behaviors with a bounded number of agents are considered. In an ex-ante Bayesian $k$-strong equilibrium, no group with at most $k$ agents can deviate from the equilibrium strategy so that every group member

gets a higher ex-ante expected utility, i.e., the expected utility before agents know their types. In a Bayesian $k$-strong equilibrium, no group with at most $k$ agents can deviate from the equilibrium strategy so that every group member gets a higher expected utility conditioned on every type.

The difference between the two solution concepts is how agents calculated their expected utilities when contemplating whether a deviation is beneficial. We interpret this difference as different attitudes of agents towards deviations. In the ex-ante Bayesian $k$-strong equilibrium, a group of agents deviates once the deviation is profitable in the ex-ante expectation. In the Bayesian $k$-strong equilibrium, an agent is assumed to be more risk-averse towards deviation and will deviate only when the deviation is profitable conditioned on every possible type. Proposition 1 shows that an ex-ante Bayesian $k$-strong equilibrium implies a Bayesian $k$-strong equilibrium.

Our technical contributions lie in the study of the collusion problem in peer prediction mechanisms, as a representative of group strategic behavior in Bayesian games, with our solution concepts. We exactly characterize the group sizes $k$ where the truthful reporting in the peer prediction mechanism by [45] is an ex-ante Bayesian $k$-strong equilibrium (Theorem 1) and a Bayesian $k$-strong equilibrium (Theorem 2), respectively. In each case, we show a threshold so that group sizes below this threshold cannot benefit by deviating while group sizes above this threshold can. In general, these thresholds are different for the two types of equilibria we consider. Our thresholds are characterized by the parameters of the game, including the number of agents, common prior, and the scoring adopted by the mechanism. Our result implies that our equilibria parameterized by $k$ are natural criteria to evaluate the robustness against collusion for a peer prediction mechanism. If truth-telling is an equilibrium with a larger $k$, the mechanism is more robust against collusion. In the application of the peer prediction mechanism, the scoring rule and the mechanism that maximizes the threshold could be chosen to prevent a wider range of collusion.

We also discuss two other possible scenarios where our solution concept may apply. In the voting scenario where voters only have partial information about the alternatives, it is known that strong equilibria with unlimited coalition sizes may fail to exist when there is a sufficiently large group of voters whose preferences are not aligned with the rest of the voters [17]. However, the sizes of the deviating groups are typically large in those non-equilibria. Given that it is unlikely for large numbers of voters to collaborate in large elections, it is therefore appealing to study equilibria with bounded deviating groups and obtain more informative results. In the private Blotto game [19], social media users with noisy information choose to annotate for/against one of the multiple posts. Agents aim to maximize the overall influence of their type on the posts. Our notion interpolates the centralized Colonel Blotto game and the decentralized private Blotto game. The parameter $k$ becomes an evaluation to characterize scenarios where agents have different centralization levels, where a higher $k$ represents a higher ability for agents to coordinate and for their type.

## 1.2 Related Work

Previous work studies group strategic behavior in Bayesian games under different scenarios. Hahn and Yannelis [27] and Safronov [52] study coalitional implementation problems under an exchange economy with a strong equilibrium. Ichiishi and Idzik [33] and Ichiishi and Yamazaki [34] propose the Bayesian strong equilibrium and study its relationship with cooperative game theory. Schoenebeck and Tao [53], Han et al. [28], and Deng et al. [17] adopt an approximated version of strong equilibrium to study information aggregation and voting with incomplete information. Nevertheless, none of these works characterizes group strategic behaviors with a bounded size of the group. Guo and Yannelis [26] proposed a coalitional interim equilibrium in which the set of admissible coalitions can be arbitrarily exogenously given. Their solution concept covers a wider range of admissible coalitions than our paper. However, truthful reporting is such an equilibrium only when agents are also coalitionally truthful when they know the report of all other agents, which does not hold for most peer prediction mechanisms. In our setting, truthful reporting fails to be such an equilibrium even with a constant coalition size under mild assumptions (Appendix E). Abraham et al. [2] proposes a $k$-coalitional equilibrium where the deviators are allowed to arbitrarily share private information, which may not be applied to many real-world scenarios. For example, the organizer can randomly assign tasks or set limited response periods to prevent agents from arbitrary communication. Moreover, truthful reporting also fails to be such an equilibrium even with a constant coalition size, as signal sharing updates the deviators' beliefs and drives them to different strategies. Our ex-ante Bayesian $k$-strong equilibrium is related to the equilibrium in [17, 28, 53], and our Bayesian $k$-strong equilibrium is an extension of the Bayesian strong equilibrium in [33]. In the game with complete information, Aumann [5] propose the strong Nash equilibrium in which no group of agents has an incentive to deviate. The strong Nash equilibrium (and its variants) has been applied to study group strategic behavior in many scenarios such as congesting game [29, 32, 62], voting [7, 18, 50], and Markov game [12, 13]. Abraham et al. [1] studies $k$-coalitional strategic behavior under games with complete information. However, a strong Nash equilibrium does not apply to Bayesian games where the information is incomplete.

Our paper is also related to studying the collusion problem in peer prediction mechanisms. Because the appropriate theoretical definitions have not been available, collusion has not been studied explicitly in theoretical peer prediction work. However, many works touch on related concepts. Intuitively, equilibrium selection is related to collusion because agents can coordinate to choose an equilibrium that is bad for the mechanism. Gao et al. [23] empirically showed this to be a problem, while Gao et al. [24] shows that agents may also coordinate on a low-effort signal. The problem of equilibrium selection is exacerbated by the inevitable existence of uninformative equilibria [35, 36, 58]. Many papers address the problem by developing mechanisms where truthful reporting is more profitable than uninformative collusions [16, 35, 36, 39, 48, 51, 60]. More powerfully, works have shown that the truthful equilibrium has the highest possible payments either among all equilibrium [38] or even among all strategies profiles [40, 55, 63]. However, all the

latter results consider multi-task peer prediction, while no single-task peer prediction mechanisms have been discovered to have the same merit. Moreover, none of these works study the collusion problem from the perspective of strong equilibrium. Schoenebeck et al. [54] studied a more extreme case where the goal was to design peer prediction mechanisms that are robust against an adversary that controls a constant fraction of the nodes. The present work is different because the deviating groups are required to be strategic and not purely malicious.

Several works study collusion using simulations and measuring how many agents must deviate before truth-telling fails to be the best response for the remaining agents [11, 57] or so that certain dynamics fail to converge back to truth-telling [56]. This shows that while the problem is interesting, the theoretical tools available for prior work were insufficient.

## 2 Preliminaries

For an integer $n$, let $[n]$ denote the set $\{1, 2, \cdots, n\}$. For a finite set $A$, let $|A|$ be the number of elements in $A$, and $\Delta_A$ denote the set of all distributions on $A$.

*Proper Scoring Rule.* Given a finite set $\mathcal{S}$, a scoring rule $PS : \mathcal{S} \times \Delta_{\mathcal{S}} \to \mathbb{R}$ maps an element $s \in \mathcal{S}$ and a distribution $\mathbf{q}$ on $\mathcal{S}$ to a score. A scoring rule $PS$ is *proper* if for any distributions $\mathbf{q}_1$ and $\mathbf{q}_2$, $\mathbb{E}_{s \sim \mathbf{q}_1}[PS(s, \mathbf{q}_1)] \geq \mathbb{E}_{s \sim \mathbf{q}_1}[PS(s, \mathbf{q}_2)]$ and *strictly proper* if the equality holds only at $\mathbf{q}_1 = \mathbf{q}_2$.

**Example 2.** *Given a distribution $\mathbf{q}$ on a finite set $\mathcal{S}$, let $q(s)$ be the probability of $s \in \mathcal{S}$ in $\mathbf{q}$. The log score rule $PS_L(s, \mathbf{q}) = \log(q(s))$. The Brier/quadratic scoring rule $PS_B(s, \mathbf{q}) = 2 \cdot q(s) - \mathbf{q} \cdot \mathbf{q}$. Both the log scoring rule and the Brier scoring rule are strictly proper.*

### 2.1 Bayesian Game Model

A Bayesian game $\mathcal{I} = ([n], (\mathcal{A}_i)_{i \in [n]}, (\mathcal{S}_i)_{i \in [n]}, (v_i)_{i \in [n]}, \mathbf{q})$ is defined by the following components.

- The set of agents $[n]$.
- For each agent $i$, $\mathcal{A}_i$ is the set of available actions of $i$. The action profile $A = (a_1, a_2, \cdots, a_n)$ is the vector of actions of all the agents.
- For each agent $i$, $\mathcal{S}_i$ is the set of possible types of agent $i$. The type characterizes the private information agent $i$ holds, and the agent can only observe his/her type in the game. The type vector $S = (s_1, s_2, \cdots, s_n)$ is the vector of types of all agents.
- For each agent $i$, $v_i : \mathcal{S}_i \times \mathcal{A}_1 \times \cdots \times \mathcal{A}_n \to \mathbb{R}$ is $i$'s utility function that maps $i$'s type and the action of all the agents to $i$'s utility.
- A *common prior* that the types of the agents follow is a joint distribution $\mathbf{q}$. For a signal $s_i$ of agent $i$, we use $q(s_i)$ to denote the marginal prior probability that $i$'s signal is $s_i$. We assume that $q(s_i) > 0$ for any $i$ and any $s_i \in \mathcal{S}_i$.

For each agent $i$, a (mixed) strategy $\sigma_i : \mathcal{S}_i \to \Delta_{\mathcal{A}_i}$ maps $i$ private signal to a distribution on his/her actions. A strategy profile $\Sigma = (\sigma_i)_{i \in [n]}$ is a vector of the strategies of all the agents.

Given a strategy profile $\Sigma$, the *ex-ante* expected utility of agent $i$ is

$$u_i(\Sigma) = \mathbb{E}_{S \sim \mathbf{q}} \mathbb{E}_A [v_i(s_i, a_1, \cdots, a_n) \mid \Sigma].$$

Similarly, given a strategy profile $\Sigma$ and a type $s_i$, the *interim* expected utility of agent $i$ conditioned on his/her type being $s_i$ is

$$u_i(\Sigma \mid s_i) = \mathbb{E}_{S_{-i} \sim \mathbf{q}_{-i|s_i}} \mathbb{E}_A [v_i(s_i, a_1, \cdots, a_n) \mid \Sigma],$$

where $S_{-i}$ is the type vector of all agents except for agent $i$, and $\mathbf{q}_{-i|s_i}$ is the joint distribution on $S_{-i}$ conditioned on agent $i$'s signal being $s_i$.

### 2.2 (Ex-ante) Bayesian $k$-Strong Equilibrium

In this paper, we focus on agents that coordinate for strategic behaviors before they know their types. This assumption relates to various constraints in real-world scenarios that prevent agents from discussions after knowing their types.

**Example 3.** *Consider the online crowdsourcing group in Example 1. The website requires workers to make an immediate report after seeing the task so that workers cannot communicate with each other after they know their types. (For example, workers have to submit the report in 30 seconds to reflect their intuition.) However, workers may collude on the same report before seeing the task.*

Both equilibria share the same high-level form: there does not exist a group of $k$ agents and a deviating strategy such that all the deviators' expected utility in the deviation is as good as the equilibrium strategy profile and at least one deviator's expected utility strictly increases. The difference lies in the expected utility. Ex-ante Bayesian $k$-strong equilibrium adopts ex-ante expected utility, while Bayesian $k$-strong equilibrium adopts interim expected utility on every type.

**Definition 1** (ex-ante Bayesian $k$-strong equilibrium). Given an integer $k \geq 1$, a strategy profile $\Sigma$ is an ex-ante Bayesian $k$-strong equilibrium ($k$-EBSE) if there does not exist a group of agent $D$ with $|D| \leq k$ and a different strategy profile $\Sigma' = (\sigma'_i)$ such that

(1) for all agent $i \notin D$, $\sigma'_i = \sigma_i$;
(2) for all $i \in D$, $u_i(\Sigma') \geq u_i(\Sigma)$;
(3) there exists an $i \in D$ such that $u_i(\Sigma') > u_i(\Sigma)$.

**Definition 2** (Bayesian $k$-strong equilibrium). Given an integer $k \geq 1$, a strategy profile $\Sigma$ is a Bayesian $k$-strong equilibrium ($k$-BSE) if there does not exist a group of agent $D$ with $|D| \leq k$ and a different strategy profile $\Sigma' = (\sigma'_i)$ such that

(1) for all agent $i \notin D$, $\sigma'_i = \sigma_i$;
(2) for every $i \in D$ and every $s_i \in \mathcal{S}_i$, $u_i(\Sigma' \mid s_i) \geq u_i(\Sigma \mid s_i)$;
(3) there exist an $i \in D$ and an $s_i \in \mathcal{S}_i$ such that $u_i(\Sigma' \mid s_i) > u_i(\Sigma \mid s_i)$.

In both solution concepts, if such a deviating group $D$ and a strategy profile $\Sigma'$ exist, we say that the deviation *succeeds*.

When $k = 1$, both ex-ante Bayesian 1-strong equilibrium and Bayesian 1-strong equilibrium are equivalent to the Bayesian Nash equilibrium [30]. (See Appendix B.) However, the two solution concepts are not equivalent for larger $k$. Example 5 illustrates a scenario in the peer prediction mechanism where the same deviation succeeds under the ex-ante Bayesian $k$-strong equilibrium but fails under the Bayesian $k$-strong equilibrium.

We interpret the difference between the two solution concepts as different attitudes of agents towards deviations. Agents are assumed to be more risk-averse towards deviations under Bayesian

$k$-strong equilibrium, as they will deviate only when the deviation brings them higher interim expected utility conditioned on every type. On the other hand, agents under the ex-ante Bayesian $k$-strong equilibrium will deviate once their ex-ante expected utility increases. Proposition 1 supports our interpretation by revealing that an ex-ante Bayesian $k$-strong equilibrium implies a Bayesian $k$-strong equilibrium.

**Proposition 1.** *For every strategy profile $\Sigma$ and every $1 \le k \le n$, if $\Sigma$ is an ex-ante Bayesian $k$-strong equilibrium, then $\Sigma$ is a Bayesian $k$-strong equilibrium.*

PROOF. Suppose $\Sigma'$ is an arbitrary deviating profile from $\Sigma$ with no more than $k$ deviators, and $i$ is an arbitrary deviator in $\Sigma'$. Since $\Sigma$ is an ex-ante Bayesian $k$-strong equilibrium, then $u_i(\Sigma') \le u_i(\Sigma)$. By the law of total probability, $u_i(\Sigma) = \sum_{s_i \in \mathcal{S}_i} \mathbf{q}(s_i) \cdot u_i(\Sigma \mid s_i)$. Therefore, one of the following must hold: (1) for all $s \in \mathcal{S}_i$, $u_i(\Sigma' \mid s_i) = u_i(\Sigma \mid s_i)$, or (2) there exists a $s \in \mathcal{S}_i$, $u_i(\Sigma' \mid s_i) < u_i(\Sigma \mid s_i)$. In either case, the deviation fails. Therefore, $\Sigma$ is a Bayesian $k$-strong equilibrium. $\square$

## 2.3 Peer Prediction Mechanism

In a peer prediction mechanism, each agent receives a private signal in $\mathcal{S} = \{\ell, h\}$ and reports it to the mechanism. All the agents share the same type set $\mathcal{S}_i = \mathcal{S}$ and action set $\mathcal{A}_i = \mathcal{S}$.

$\mathbf{q}$ is the common prior joint distribution of the signals. Let $\Psi_i$ denote the random variable of agent $i$'s private signal. We assume that the common prior $\mathbf{q}$ is symmetric — for any permutation $\pi$ on $[n]$, $\mathbf{q}(\Psi_1 = s_1, \Psi_2 = s_2, \cdots, \Psi_n = s_n) = q(\Psi_1 = s_{\pi(1)}, \Psi_2 = s_{\pi(2)}, \cdots, \Psi_n = s_{\pi(n)})$.

$q(s)$ is the prior marginal belief that an agent has signal $s$, and $q(s \mid s')$ be the posterior belief of an agent with private signal $s'$ on another agent having signal $s$. We also define $\mathbf{q}_s = q(\cdot \mid s)$ be the marginal distribution on $\mathcal{S}$ conditioned on $s$. We assume that an agent with $h$ signal has a higher estimation than an agent with $\ell$ signal on the probability that another agent has $h$ signal, i.e., $q(h \mid h) > q(h \mid \ell)$. We also assume that any pair of signals is not fully correlated, which is $q(h \mid \ell) > 0$ and $q(\ell \mid h) > 0$.

We adopt a modified version of the peer prediction mechanism [45] characterized by a (strictly) proper scoring rule $PS$. The mechanism compares the report of agent $i$, denoted by $a_i$, with the reports of all other agents. For each agent $j$ with report $a_j$, the reward $i$ gains from comparison with $j$'s report is $R_i(a_j) = PS(a_j, \mathbf{q}_{a_i})$. The utility of agent $i$ is the average reward from each $j$.

$$v_i(s_i, A) = \frac{1}{n-1} \sum_{j \in [n], j \ne i} R_i(a_j).$$

**Remark 1.** *In the original mechanism in [45], the reward of an agent $i$ is $R_i(a_j)$, where $j$ is chosen uniformly at random from all other agents. We derandomize the mechanism so that it fits better into the Bayesian game framework while the expected utility of an agent is unchanged.*

**Example 4.** *Suppose $n = 100$. For the common prior, the prior belief $q(h) = 2/3$, and $q(\ell) = 1/3$. The posterior belief $q(h \mid h) = 0.8$ and $q(\ell \mid \ell) = 0.6$. Suppose the Brier scoring rule is applied to the peer prediction mechanism. Consider an agent $i$ with report $a_i = h$. Then,*

*$i$'s reward from a peer $j$ with report $a_j = h$ is $R_i(a_j) = PS_B(h, \mathbf{q}_h) = 2 \cdot q(h \mid h) - q(h \mid h)^2 - q(\ell \mid h)^2 = 0.92$. Similarly, $i$' reward from another peer $j'$ with report $a_{j'} = \ell$ is $PS_B(\ell, \mathbf{q}_h) = -0.28$.*

A (mixed) strategy $\sigma : \mathcal{S}_i \to \Delta_{\mathcal{A}_i}$ maps an agent's type to a distribution on his/her action. A strategy profile $\Sigma = (\sigma_i)_{i \in [n]}$ is a vector of the strategies of all the agents. An agent is *truthful* if he/she always truthfully reports his/her private signal. Let $\sigma^*$ be the truthful strategy and $\Sigma^*$ be the strategy profile where all agents are truthful. We also represent a strategy in the form $\sigma = (\beta_\ell, \beta_h) \in [0, 1]^2$, where $\beta_\ell$ and $\beta_h$ are the probability that an agent playing $\sigma$ reports $h$ conditioned on his/her signal begin $\ell$ and $h$, respectively. The truthful strategy $\sigma^* = (0, 1)$.

Given the strategy profile $\Sigma$, the ex-ante expected utility of an agent $i$ is

$$u_i(\Sigma) = \frac{1}{n-1} \sum_{j \in [n], j \ne i} \mathbb{E}_{s_i \sim \mathbf{q}, a_i \sim \sigma_i(s_i)} \mathbb{E}_{s_j \sim \mathbf{q}_{s_i}, a_j \sim \sigma_j(s_j)} R_i(a_j).$$

Given a strategy profile $\Sigma$ and a type $s_i$, the interim expected utility of an agent $i$ conditioned on his/her type being $s_i$ is

$$u_i(\Sigma \mid s_i) = \frac{1}{n-1} \sum_{j \in [n], j \ne i} \mathbb{E}_{a_i \sim \sigma_i(s_i)} \mathbb{E}_{s_j \sim \mathbf{q}_{s_i}, a_j \sim \sigma_j(s_j)} R_i(a_j).$$

**Example 5.** *We follow the setting in example 4. Let $\Sigma^*$ be the profile where all agents report truthfully. Let $D$ be a group containing $k = 40$ agents and $\Sigma'$ be the profile where all deviators report $h$.*

*For truthful reporting, consider an agent $i$ and his/her peer $j$. The probability that both $i$ and $j$ receive (and report) signal $h$ is $q(h) \cdot q(h \mid h) = 2/3 * 0.8 = 0.533$, and $i$ will be rewarded $PS(h, \mathbf{q}_h) = 0.92$. Other probabilities can be calculated similarly. Adding on the expectation of different pairs of signals, we can calculate the ex-ante expected utility of $i$ in truthful reporting: $u_i(\Sigma^*) = \sum_{s_i, s_j \in \{\ell, h\}} q(s_i) \cdot q(s_j \mid s_i) \cdot PS(s_j, \mathbf{q}_{s_i}) = 0.627$.*

*Now we consider the expected utility of a deviator $i$ deviating profile $\Sigma'$. Since all the deviators always report $h$, the expected reward $i$ gets from a deviator is $PS(h, \mathbf{q}_h) = 0.92$. For the rewards from a truthful reporter, $i$'s expected reward is $q(h) \cdot PS(h, \mathbf{q}_h) + q(\ell) \cdot PS(\ell, \mathbf{q}_h) = 0.52$. Among all the other agents, $k - 1 = 39$ agents are deviators, and $n - k = 60$ agents are truthful reporters. Therefore, $i$'s expected utility on $\Sigma'$ is $u_i(\Sigma') = 0.682 > u_i(\Sigma^*)$. Therefore, the deviation succeeds under the ex-ante Bayesian $k$-strong equilibrium.*

*However, the deviation fails under the Bayesian $k$-strong equilibrium. The truthful expected utility conditioned on $i$'s signal is $\ell$ is $u_i(\Sigma^* \mid \ell) = \sum_{s_j \in \{\ell, h\}} q(s_j \mid \ell) \cdot PS(s_j, \mathbf{q}_\ell) = 0.52$. On the other hand, when agents deviate to $\Sigma'$, $i$'s reward from a truthful agents becomes $\sum_{s_j \in \{\ell, h\}} q(s_j \mid \ell) \cdot PS(s_j, \mathbf{q}_h) = 0.2$. Therefore, $i$'s interim expected utility $u_i(\Sigma' \mid \ell) = 0.484 < u_i(\Sigma^* \mid \ell)$.*

## 3 Dichotomies on Equilibria

Our theoretical results focus on the collusive behavior in the peer prediction mechanisms. While the mechanism is known to be prone to collusions, Shnayder and Parkes [57] empirically shows that there is a lower bound for collusion to be profitable. With our new solution concepts, our theoretical results specify the exact threshold. For both equilibria, we find the largest group size $k_E$ ($k_B$, respectively) such that truthful reporting is an equilibrium. Moreover, for any $k$ larger than $k_E$ ($k_B$, respectively), truthful reporting fails to be

an equilibrium. We first present the result of the ex-ante Bayesian $k$-strong equilibrium.

**Theorem 1.** *In the peer prediction mechanism, for any $n \geq 2$ and any strictly proper scoring rule PS, truthful reporting $\Sigma^*$ is an ex-ante Bayesian $k_E$-strong equilibrium, where*

$$k_E^h = \begin{cases} \left\lfloor \frac{(n-1) \cdot \mathbb{E}_{s \sim \mathbf{q}_\ell}[PS(s, \mathbf{q}_\ell) - PS(s, \mathbf{q}_h)]}{PS(h, \mathbf{q}_h) - PS(\ell, \mathbf{q}_h)} \right\rfloor + 1 & \text{if } PS(h, \mathbf{q}_h) > PS(\ell, \mathbf{q}_h) \\ n & \text{otherwise} \end{cases}$$

$$k_E^\ell = \begin{cases} \left\lfloor \frac{(n-1) \cdot \mathbb{E}_{s \sim \mathbf{q}_h}[PS(s, \mathbf{q}_h) - PS(s, \mathbf{q}_\ell)]}{PS(\ell, \mathbf{q}_\ell) - PS(h, \mathbf{q}_\ell)} \right\rfloor + 1 & \text{if } PS(\ell, \mathbf{q}_\ell) > PS(h, \mathbf{q}_\ell) \\ n & \text{otherwise} \end{cases}$$

$$k_E = \min(k_E^h, k_E^\ell, n).$$

*For all $n \geq k > k_E$, truthful reporting is NOT an ex-ante Bayesian $k$-strong equilibrium.*

While a proof sketch is presented below, here we give a brief explanation of the thresholds. $k_E^h$ and $k_E^\ell$ are characterized by comparing the ex-ante expected utility of a deviator between truthful reporting and all the deviators always report $h$ ($\ell$, respectively). Take $k_E^h$ as an example. The numerator $\mathbb{E}_{s \sim \mathbf{q}_\ell}[PS(s, \mathbf{q}_\ell) - PS(s, \mathbf{q}_h)]$ is proportional to the loss that the deviator suffers in his expected rewards from the truthful reporters in switching from truthful reporting to always reporting $h$. The denominator $PS(h, \mathbf{q}_h) - PS(\ell, \mathbf{q}_h)$ is proportional to the amount that, for a deviator, the expected reward gain from other deviators exceeds the expected reward loss from truthful reporting. If $PS(h, \mathbf{q}_h) - PS(\ell, \mathbf{q}_h) < 0$, the extra gain never compensates for the loss, so the deviation cannot succeed for any $k \leq n$. Otherwise, a group size of $k > k_E^h$ is required for the deviation to succeed.

**Example 6.** *We calculate the threshold for ex-ante Bayesian $k$-strong equilibrium for the instance in Example 4. For $k_E^h$, the numerator equals to $q(h \mid \ell) \cdot (0.28 - 0.92) + q(\ell \mid \ell) \cdot (0.68 + 0.28) = 0.32$. The denominator, according to the Brier scoring rule, equals to $PS(h, \mathbf{q}_h) - PS(\ell, \mathbf{q}_h) = 2 \cdot (q(h \mid h) - q(\ell \mid h)) = 1.2$. Therefore, $k_E^h = \lfloor \frac{4}{15} \cdot (n - 1) \rfloor + 1$. Similarly, we calculate that $k_E^\ell = \lfloor \frac{4}{5} \cdot (n - 1) \rfloor + 1$. Therefore, when $n = 100$, a deviation group needs at least $\lfloor \frac{4}{15} \times 99 \rfloor + 1 = 27$ deviators to succeed. This aligns with Example 5, where a 40-agent group succeeds.*

Similarly, Theorem 2 characterizes the threshold under Bayesian $k$-strong equilibrium.

**Theorem 2.** *In the peer prediction mechanism, there exists an $n_0$ such that for every $n \geq n_0$ and any strictly proper scoring rule PS, truthful reporting $\Sigma^*$ is a Bayesian $k$-strong equilibrium in peer prediction, where*

$$k_B^h = \begin{cases} \left\lceil \frac{(n-1) \cdot \mathbb{E}_{s \sim \mathbf{q}_\ell}[PS(s, \mathbf{q}_\ell) - PS(s, \mathbf{q}_h)]}{q(\ell \mid \ell) \cdot (PS(h, \mathbf{q}_h) - PS(\ell, \mathbf{q}_h))} \right\rceil & \text{if } PS(h, \mathbf{q}_h) > PS(\ell, \mathbf{q}_h) \\ n & \text{otherwise} \end{cases}$$

$$k_B^\ell = \begin{cases} \left\lceil \frac{(n-1) \cdot \mathbb{E}_{s \sim \mathbf{q}_h}[PS(s, \mathbf{q}_h) - PS(s, \mathbf{q}_\ell)]}{q(h \mid h) \cdot (PS(\ell, \mathbf{q}_\ell) - PS(h, \mathbf{q}_\ell))} \right\rceil & \text{if } PS(\ell, \mathbf{q}_\ell) > PS(h, \mathbf{q}_\ell) \\ n & \text{otherwise} \end{cases}$$

$$k_B = \min(k_B^h, k_B^\ell, n).$$

*For all $n \geq k > k_B$, truthful reporting is NOT a Bayesian $k$-strong equilibrium.*

The lower bound $n_0$ on $n$ is characterized by the common prior $q$ and the scoring rule $PS$ and is independent of $n$. The explicit expression on $n_0$ is in Appendix D.

The thresholds for the Bayesian $k$-strong equilibrium $k_B^h$ and $k_B^\ell$ are larger than those for the ex-ante Bayesian $k$-strong equilibrium $k_E^h$ and $k_E^\ell$ respectively. This is because, for example, $k_B^h$ is characterized by comparing the interim utility of a deviator conditioned on signal $\ell$ between truthful reporting and all deviators reporting $h$. In ex-ante, the deviator $i$ has a probability of $q(\ell)$ to report untruthfully and suffer a loss on expected reward from truthful reporters. When $i$ has a private signal $\ell$, such probability becomes 1. Therefore, the deviator suffers more loss in the interim expected utility than in the ex-ante expected utility in the expected reward from truthful reporters. On the other hand, $i$ gets the same extra gain in the reward from other deviators as in the ex-ante expected utility. Therefore, a larger group is needed to make the deviation succeed.

**Example 7.** *We calculate the threshold for Bayesian $k$-strong equilibrium for the instance in Example 4. For $k_B^h$, the numerator equals to 0.32. The denominator is multiplied by $q(\ell \mid \ell) = 0.6$ compared with $k_E^h$ and equals to $1.2 \times 0.6 = 0.72$. Therefore, $k_E^h = \lceil \frac{4}{9} \cdot (n - 1) \rceil$. Similarly, we calculate that $k_E^\ell = \lceil n - 1 \rceil$. Therefore, when $n = 100$, a deviation group needs at least 45 deviators to succeed. This also aligns with Example 5, where a 40-agent group fails in deviation.*

Theorem 1 and 2 imply that the ex-ante Bayesian $k$-strong equilibrium and the Bayesian $k$-strong equilibrium are natural criteria to evaluate the robustness against collusion for a peer prediction mechanism. If truth-telling is an equilibrium with a larger $k$, the mechanism is more robust against collusion. If an information collector aims to prevent collusion in a peer prediction task, he/she could carefully select the mechanism and the scoring rule to maximize the threshold $k$ under which truth-telling becomes an equilibrium.

**Example 8.** *If we change the scoring rule from the Brier scoring rule Example 4 to the log scoring rule with base $e$ in and follow the calculation in Example 6, we have $k_E = \lfloor 0.275(n - 1) \rfloor + 1$. When $n = 100$, a group of at least 28 agents is needed to perform a successful deviation. Therefore, the log scoring rule is more robust than the Brier scoring rule in this instance.*

### 3.1 Proof Sketch of Theorem 1

The proof consists of two steps. In Step 1, $k_E$ is characterized by comparing the ex-ante expected utility of a deviator when every agent reports truthfully and when all $k$ deviators always report $h$ (and always report $\ell$, respectively). The two deviations bring a deviator higher expected utility if and only if $k > k_E$. In Step 2, we show that for any $k \leq k_E$ and any deviating strategy profile $\Sigma'$, the average expected utility among all the deviators when $\Sigma'$ is played will not exceed the expected utility when every agent reports truthfully. Therefore, either no deviators have strictly increasing expected utility or some deviators have strictly decreasing utility after deviation, and the deviation cannot succeed. The full proof is in Appendix C.

**Step 1: determine $k_E$.** We show how $k_E^h$ is determined by comparing truthful reporting strategy profile $\Sigma^*$ and the deviating strategy

profile $\Sigma$ where all $k$ deviators always report $h$, i.e., $\sigma = (1, 1)$. The reasoning for $k_E^\ell$ is similar. The condition that a deviator $i$ is willing to deviate is $u_i(\Sigma) > u_i(\Sigma^*)$. The inequality should be strict because all deviators have equal expected utility in $\Sigma$.

$u_i(\Sigma)$ can be viewed as a linear combination of the expected utility $i$ gets from the truthful agents, denoted by $u_i(\Sigma \mid \text{truthful})$, and the expected utility $i$ gets from other deviators, denoted by $u_i(\Sigma \mid \text{deviator})$. In $\Sigma$, there are $n - k$ truthful reporters and $k - 1$ deviators other than $i$. Therefore, $u_i(\Sigma) = \frac{n-k}{n-1} \cdot u_i(\Sigma \mid \text{truthful}) + \frac{k-1}{n-1} \cdot u_i(\Sigma \mid \text{deviator})$. Let $\Delta u_d = u_i(\Sigma^*) - u_i(\Sigma \mid \text{deviator})$, and $\Delta u_t = u_i(\Sigma^*) - u_i(\Sigma \mid \text{truthful})$. Then $u_i(\Sigma) > u_i(\Sigma^*)$ is equivalent to $\frac{k-1}{n-1} \cdot \Delta u_d + \frac{n-k}{n-1} \cdot \Delta u_t < 0$.

The ex-ante expected reward of deviator $i$ from truthful reporters can be divided into two parts, one conditioned on $i$'s private signal being $h$, the other on $i$'s signal being $\ell$. When $i$'s signal is $h$, $i$ reports $h$ both in $\Sigma^*$ and in $\Sigma$, and the expected rewards from truthful reporters in this part are the same. When $i$'s signal is $\ell$, $i$ reports $\ell$ in $\Sigma^*$ and $h$ in $\Sigma$, and the expected rewards make a difference. Therefore, $\Delta u_t = q(\ell) \cdot \mathbb{E}_{s \sim \mathbf{q}_\ell}[PS(s, \mathbf{q}_\ell) - PS(s, \mathbf{q}_h)]$. According to the properness of $PS$, $\Delta u_t > 0$. Therefore, when $\Delta u_t > \Delta u_d$, $u_i(\Sigma) > u_i(\Sigma^*)$ is equivalent to

$$k > \frac{\Delta u_t}{\Delta u_t - \Delta u_d} \cdot (n - 1) + 1.$$

When $\Delta u_t \leq \Delta u_d$, the condition does not hold for any $k$, and the deviation will never succeed.

From the calculation, $\Delta u_t - \Delta u_d = q(\ell) \cdot (PS(h, \mathbf{q}_h) - PS(\ell, \mathbf{q}_h))$. Therefore, when $PS(h, \mathbf{q}_h) > PS(\ell, \mathbf{q}_h)$, $\Delta u_t > \Delta u_d$, and $u_i(\Sigma) > u_i(\Sigma^*)$ is equivalent to

$$k > \frac{\mathbb{E}_{s \sim \mathbf{q}_\ell}[PS(s, \mathbf{q}_\ell) - PS(s, \mathbf{q}_h)]}{PS(h, \mathbf{q}_h) > PS(\ell, \mathbf{q}_h)} \cdot (n - 1) + 1.$$

And when $PS(h, \mathbf{q}_h) \leq PS(\ell, \mathbf{q}_h)$, $\Delta u_t \leq \Delta u_d$, and $u_i(\Sigma) > u_i(\Sigma^*)$ does not hold for any $k$. This is how $k_E^h$ is determined. $k_E^\ell$ is determined in a similar reasoning.

**Step 2: Deviations cannot succeed for $k \leq k_E$.** For $k = 1$, the statement holds from the truthfulness of the mechanism. Suppose $2 \leq k \leq k_E$, and $\Sigma$ be an arbitrary deviating strategy. Let $u(\Sigma^*)$ be the expected utility of truthful reporting, which is equal for all agents. For each deviator $i$, let $\sigma_i = (\beta_\ell^i, \beta_h^i)$ denote $i$'s strategy in $\Sigma$. We show that $\frac{1}{k} \sum_{i \in D} u_i(\Sigma) \leq u(\Sigma^*)$. Therefore, either there exists some deviator $i$ such that $u_i(\Sigma) < u(\Sigma^*)$, or for all the deviator $i$ there is $u_i(\Sigma) = u(\Sigma^*)$. In either case, the deviation fails.

Now let $\bar{\sigma} = (\bar{\beta}_\ell, \bar{\beta}_h) = \frac{1}{k} \sum_{i \in D} \sigma_i$ be the average of the deviator's strategies, and $\bar{\Sigma}$ be the strategy profile where all agents in $D$ plays $\bar{\sigma}$ and all other agents report truthfully. $u_i(\bar{\Sigma})$ is equal among all the deviators $i$ due to symmetricity (and denoted by $u(\bar{\Sigma})$). We first show that $\frac{1}{k} \sum_{i \in D} u_i(\Sigma) \leq u(\bar{\Sigma})$ (the average expected utility of deviators playing $\Sigma$ will not exceed the expected utility when each deviator plays $\bar{\sigma}$) and then that $u(\bar{\Sigma}) \leq u(\Sigma^*)$ (the expected utility that each deviator play $\bar{\sigma}$ will not exceed the truthful expected utility).

To show $\frac{1}{k} \sum_{i \in D} u_i(\Sigma) \leq u(\bar{\Sigma})$, we compare the expected reward from truthful agents and deviators separately. For a deviator $i$, $u_i(\Sigma \mid \text{truthful})$ is independent of the strategy of other deviators and is linear on $\beta_\ell$ and $\beta_h$. Therefore, $\frac{1}{k} \sum_{i \in D} u_i(\Sigma \mid \text{truthful}) = u(\bar{\Sigma} \mid \text{truthful})$.

For the deviator's part, $u_i(\Sigma \mid \text{deviator})$ is the average of $i$'s expected reward from comparing the report with all other deviators $j \in D$. Such expected reward is linear on $j$'s strategy given a fixed $i$'s strategy and linear on $i$'s strategy given a fixed $j$'s strategy. Therefore, the average expected reward from agents with different strategies equals to the reward from a peer playing the average strategy, and $u_i(\Sigma \mid \text{deviator})$ equals to $i$'s expected reward from an agent playing the average strategy $\bar{\sigma}$ minus a share of $i$'s expected reward from an agent playing $\sigma_i$. Given a strategy $\sigma = (\beta_\ell, \beta_h)$, let $f(\beta_\ell, \beta_h)$ be the expected reward of an agent playing $\sigma$ from another agents also playing $\sigma$. Then

$$\frac{1}{k} \sum_{i \in D} u_i(\Sigma \mid \text{deviator}) = \frac{k}{k-1} f(\bar{\beta}_\ell, \bar{\beta}_h) - \frac{1}{(k-1)k} \sum_{i \in D} f(\beta_\ell^i, \beta_h^i).$$

It turns out that $f$ is a convex function. Therefore, $\frac{1}{k} \sum_{i \in D} u_i(\Sigma \mid \text{deviator}) \leq f(\bar{\beta}_\ell, \bar{\beta}_h) = u(\bar{\Sigma} \mid \text{deviator})$. Combining the truthful part and the deviator part, we show that $\frac{1}{k} \sum_{i \in D} u_i(\Sigma) \leq u(\bar{\Sigma})$.

Finally, we show that $u(\bar{\Sigma}) \leq u(\Sigma^*)$. Note that $u(\bar{\Sigma})$ can be viewed as a convex function on $\bar{\beta}_\ell$ and $\bar{\beta}_h$. This is because $u(\bar{\Sigma} \mid \text{truthful})$ is linear on $\bar{\sigma}$, and $u(\bar{\Sigma} \mid \text{deviator}) = f(\bar{\beta}_\ell, \bar{\beta}_h)$ is convex on $\bar{\beta}_\ell$ and $\bar{\beta}_h$. Therefore, it is sufficient to show that $u(\bar{\Sigma}) \leq u(\Sigma^*)$ on the four corner cases of $\bar{\sigma}$: truthful reporting: $\bar{\sigma} = (0, 1)$, always reporting $h$: $\bar{\sigma}(1, 1)$, always reporting $\ell$: $\bar{\sigma} = (0, 0)$, and always tell a lie $\bar{\sigma} = (1, 0)$. When $\bar{\sigma} = (0, 1)$, all the deviator also report truthfully, and $\bar{\Sigma} = \Sigma^*$. For $\bar{\sigma} = (0, 0)$ and $\bar{\sigma} = (1, 1)$, $k \leq k_E$ guarantees that $u(\bar{\Sigma}) \leq u(\Sigma^*)$. Finally, when $\bar{\sigma} = (1, 0)$, similar reasoning to Step 1 shows that such deviation cannot succeed. $\square$

## 3.2 Proof Sketch of Theorem 2.

The steps of the proof resemble the steps of the proof of Theorem 1, yet the techniques are different. In Step 1, we determine $k_B$ by comparing the interim expected utilities of a deviator when every agent reports truthfully and when all $k$ deviators always report $h$ ($\ell$, respectively). In Step 2, we show that for any $k \leq k_E$ and any deviating strategy profile $\bar{\Sigma}$ where all the deviators play the same strategy $\bar{\sigma}$, the expected utility of a deviator on $\bar{\Sigma}$ will not exceed the expected utility when every agent reports truthfully. In Step 3, we show that for sufficiently large $n$, any $k \leq k_B$, and any deviating strategy profile $\Sigma$, there exists a deviator whose expected utility is strictly smaller than the expected utility when every agent reports truthfully. The full proof is in Appendix D.

The main technical difficulty lies in Step 2 and Step 3. Let $u(\bar{\Sigma} \mid h)$ and $u(\bar{\Sigma} \mid \ell)$ be the interim expected utility of a deviator conditioned on his/her signal being $h$ and $\ell$, respectively, when $\bar{\Sigma}$ is played. $u(\bar{\Sigma} \mid h)$ and $u(\bar{\Sigma} \mid \ell)$ can still be viewed as functions on $\bar{\beta}_\ell$ and $\bar{\beta}_h$. However, unlike the ex-ante $u(\bar{\Sigma})$, they are not convex. Therefore, we cannot get $\frac{1}{k} \sum_{i \in D} u_i(\Sigma \mid h) \leq u(\bar{\Sigma} \mid h)$ or $u(\bar{\Sigma} \mid h) \leq u(\Sigma^* \mid h)$ (or the $\ell$ side) directly from similar reasoning with those in Theorem 1.

**In Step 2**, we instead show that for any $\bar{\Sigma}$, either $u(\bar{\Sigma} \mid h) \leq u(\Sigma^* \mid h)$ or $u(\bar{\Sigma} \mid \ell) \leq u(\Sigma^* \mid \ell)$ holds. Although $u(\bar{\Sigma} \mid h)$ is not convex, the convexity (or linearity) still holds in certain directions. Here we slightly abuse the notation to write $u(\bar{\Sigma} \mid h)$ as $u(\bar{\beta}_\ell, \bar{\beta}_h \mid h)$. The following properties hold. (1) When $\bar{\beta}_\ell$ is fixed, $u(\bar{\beta}_\ell, \bar{\beta}_h \mid h)$ is convex on $\bar{\beta}_h$. (2) When $\bar{\beta}_h$ is fixed, $u(\bar{\beta}_\ell, \bar{\beta}_h \mid h)$ is linear on $\bar{\beta}_\ell$. (3) When $\bar{\beta}_\ell = \frac{q(h|h)}{q(\ell|h)} \cdot (1 - \bar{\beta}_h)$, $u(\bar{\beta}_\ell, \bar{\beta}_h \mid h)$ is linear on $\bar{\beta}_h$ and

increases when $\bar{\beta}_h$ increases. With these properties, we show that $u(\bar{\beta}_\ell, \bar{\beta}_h \mid h) \le u(\Sigma^* \mid h)$ holds in a triangle area as the following lemma indicates.

**Lemma 1.** *For any $(\bar{\beta}_\ell, \bar{\beta}_h) \in \mathbb{R}^2$ satisfying (1) $\bar{\beta}_\ell \ge 0$, (2) $\bar{\beta}_h \ge 0$, and (3) $\bar{\beta}_h + \frac{q(\ell|h)}{q(h|h)} \cdot \bar{\beta}_\ell \le 1$, it always holds that $u(\bar{\beta}_\ell, \bar{\beta}_h \mid h) \le u(\Sigma^* \mid h)$, and the equality holds only when $\bar{\beta}_\ell = 0$ and $\bar{\beta}_h = 1$.*

A similar triangle characterization also applies to the $|\ell$ side.

**Lemma 2.** *For any $(\bar{\beta}_\ell, \bar{\beta}_h) \in \mathbb{R}^2$ satisfying (1) $\bar{\beta}_\ell \le 1$, (2) $\bar{\beta}_h \le 1$, and (3) $\bar{\beta}_h + \frac{q(\ell|\ell)}{q(h|\ell)} \cdot \bar{\beta}_\ell \ge 1$, it always holds that $u(\bar{\beta}_\ell, \bar{\beta}_h \mid \ell) \le u(\Sigma^* \mid \ell)$, and the equality holds only when $\bar{\beta}_\ell = 0$ and $\bar{\beta}_h = 1$.*

The union of the two triangles covers $[0, 1]^2$, as $\frac{q(h|h)}{q(h|h)} < \frac{q(\ell|\ell)}{q(h|\ell)}$. Therefore, for any $\bar{\sigma} \ne (0, 1)$, the interim expected utility of the deviators will be strictly lower than that of truthful reporting conditioned on at least one of the signals.

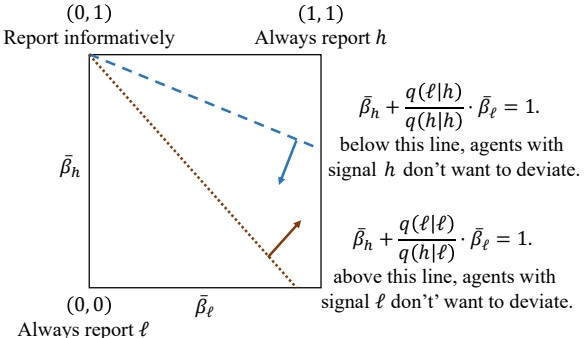

**Figure 1: The illustration of Lemma 1 and 2. The X-axis and Y-axis denote $\bar{\beta}_\ell$ and $\bar{\beta}_h$ respectively. The two half-planes characterized by two lines cover the $[0, 1]^2$ area, so there always exists agents with a certain signal that do not wish to deviate. Two lines are not necessarily located above/below point (1, 0).**

**Step 3** consists of three parts. Here we present the reasoning conditioned on private signal being $h$. The reasoning of the $\ell$ side is similar. Given a deviating strategy $\Sigma$, let $\bar{\Sigma}$ be the strategy where all the deviators play the average strategy $\bar{\sigma} = \frac{1}{k}\sum_{i \in D}\sigma_i$ in $\Sigma$. $\bar{\sigma}$ will be located in the area characterized by Lemma 1. In the first part, we show that when the deviators switch from $\bar{\Sigma}$ to $\Sigma$, there exists some agent $i$ whose expected utility will not increase by $\frac{C}{n-1}$, where $C$ is a constant related to the scoring rule. Formally, $u_i(\Sigma \mid h) \le u(\bar{\Sigma} \mid h) + \frac{C}{n-1}$. This comes from the fact that $u_i(\Sigma \mid h) - u(\bar{\Sigma} \mid h)$ can be written in the form of $u_i(\Sigma \mid h) - u(\bar{\Sigma} \mid h) = M(\bar{\Sigma}) \cdot (\beta_h^i - \bar{\beta}_h) + O(\frac{1}{n-1})$, where $M$ is a function of $\bar{\Sigma}$. Therefore, for any fixed $\bar{\Sigma}$, either an agent $i$ with $\beta_h^i \le \bar{\beta}_h$ or with $bph^i \ge \bar{\beta}_h$ satisfies $u_i(\Sigma \mid h) \le u(\bar{\Sigma} \mid h) + \frac{C}{n-1}$. In the second part, with similar reasoning for Lemma 1, we show that for sufficiently large $n$ and any $\bar{\sigma}$ in the triangle range, $u(\bar{\Sigma} \mid h) \ge u(\Sigma^* \mid h) - \frac{C}{n-1}$ only if $\bar{\sigma}$ is close to $(0, 1)$ or $(0, 0)$. In all other cases, the upper bound of the gain from switching from $\bar{\Sigma}$ to $\Sigma$ is insufficient to fill the gap between $\bar{\Sigma}$ and truth-telling $\Sigma^*$ ($u_i(\Sigma \mid h) \le u(\bar{\Sigma} \mid h) + \frac{C}{n-1} < u(\Sigma^* \mid h)$).

Finally, we show that in the corner cases, there is a deviator $i$ such that $u_i(\Sigma \mid h) < u(\Sigma^* \mid h)$. $\qquad \square$

## 4 Other Applications and Future Directions

Although our theoretical results focus on peer prediction, we believe that our solution concept of (ex-ante) Bayesian $k$-strong equilibrium is a powerful tool to characterize coalitional strategic behaviors (with bounded group size) and predict the outcome in a wide range of real-world scenarios. Here we give two more scenarios in which our solution can be applied: voting with partially informed voters and the Private Blotto game.

### 4.1 Voting with Partially Informed Voters

Starting from the Condorcet Jury Theorem [14], voting with partially informed voters has been extensively studied in the past literature [3, 4, 6, 8, 9, 15, 20–22, 25, 28, 37, 41–43, 46, 47, 59]. In the voting setting where voters are partially informed, each voter only has partial information about the alternatives and his/her preference over the alternatives is not immediately clear. This happens in myriad scenarios such as presidential elections (where the performance of each president candidate is not fully known) and voting for or against a certain policy or a certain decision (where the effect of the policy/decision is unclear at the moment of the voting). The goal of a voting scheme is to aggregate the information of the voters and uncover the alternative favored by the majority *ex-post*. This is already a non-trivial task in the information aggregation aspect [49], and the situation is even more complex with strategic agents. In fact, this problem is highly non-trivial even with two alternatives.

Consider the following typical model. A set of $n$ voters are voting between two alternatives $\{\mathcal{A}, \mathcal{B}\}$. There are two world states $\{X, Y\}$. One of them is *the actual world*, but this is unknown to the voters. A voter's preference over $\{\mathcal{A}, \mathcal{B}\}$ may or may not depend on the world state. For example, a voter may prefer $\mathcal{A}$ over $\mathcal{B}$ if the actual world is $X$ and $\mathcal{B}$ over $\mathcal{A}$ if the actual world is $Y$, while another voter may always prefer $\mathcal{A}$ over $\mathcal{B}$ regardless of the world state. The goal of a voting mechanism is to identify the alternative that is favored by the majority if the actual world state were revealed. Although voters do not know the actual world state, each voter receives a signal that is correlated to the world states. A voter, after receiving the signal, forms beliefs over the likelihood of each world state, infers the signals received by other voters, and casts a vote according to these pieces of information. This naturally formulates the problem as a Bayesian game. In addition, when the number of voters $n$ is large, an individual voter's change of strategy is unlikely to affect the outcome of the election, and thus his/her expected utility is almost unrelated to his/her strategy. On the other hand, voters form coalitions and jointly decide their votes in many practical scenarios. This motivates the study of strong Bayes-Nash equilibria.

The celebrated result from Han et al. [28] shows that, when voters' preferences are *aligned*, under the *majority vote mechanism* (each voter votes for either $\mathcal{A}$ or $\mathcal{B}$; the alternative voted by more than half of the voters wins), the alternative favored by the majority (in *ex-post*) is almost surely identified *if and only if* the strategy profile is a strong Bayes Nash equilibrium. Here, by saying aligned

preferences, we mean that all voters' utilities for alternative $\mathcal{A}$ are higher in world state $X$ than in world state $Y$ and their utilities for alternative $\mathcal{B}$ are higher in world state $Y$ than in world state $X$. That is, all voters' preferences are aligned in that they agree $X$ "corresponds to" $\mathcal{A}$ and $Y$ "corresponds to" $\mathcal{B}$, although the extent the voters' preferences are aligned with this correspondence can be different and due to which voters can be classified into three types:

- the "left-wing voters" who always prefer $\mathcal{A}$: $v(\mathcal{A}, X) > v(\mathcal{A}, Y) > v(\mathcal{B}, Y) > v(\mathcal{B}, X)$
- the "right-wing voters" who always prefer $\mathcal{B}$: $v(\mathcal{B}, Y) > v(\mathcal{B}, X) > v(\mathcal{A}, X) > v(\mathcal{A}, Y)$
- "swing voters" who prefer the alternative corresponding to the world state: $v(\mathcal{A}, X) > v(\mathcal{B}, X)$ and $v(\mathcal{B}, Y) > v(\mathcal{A}, Y)$

where $v(a, s)$ denotes the utility for alternative $a \in \{\mathcal{A}, \mathcal{B}\}$ given the actual world state $s \in \{X, Y\}$.

The story is much more complicated with general utilities $v(\cdot, \cdot)$ that are not necessarily aligned. In Deng et al. [17], it is proved that strong Bayes Nash equilibria may not exist even with only two types of voters with antagonistic preferences. In particular, "good" equilibria that identify the majority-favored alternative only exist when the voters from one type significantly outnumber the voters from the other type. When the population sizes of the two types of voters are close, Deng et al. [17] show that no strong Bayes Nash equilibrium exists.

However, only strong equilibria with unrestrictive group sizes are considered in the above-mentioned work. A typical deviation group in a strategy profile consists of all voters of the same type, and the existence of this kind of large deviation group prevents many strategy profiles from being equilibria. When considering more practical scenarios with bounded coalition sizes, more "good" equilibria are attainable. Given the large size of deviation groups in the non-equilibria found in Deng et al. [17], it is likely there is an interpolation between the deviation group size $k$ and the distribution of voters from different types where "good" equilibria exist. This provides a fine-grained structure to the problem compared with the "all-or-nothing" result in Deng et al. [17].

Strong equilibria are even less likely to exist for more general utilities. It is appealing to apply our new equilibrium concepts with bounded deviating group sizes to characterize voters' strategic behaviors and obtain more positive and fine-grained results. We believe this is a challenging yet exciting future research direction.

## 4.2 Private Blotto Game

Private Blotto game [19] is a decentralized variation of the classic Colonel Blotto game [10]. It is proposed in order to model the conflict in the crowdsourcing social media annotation. For example, the Community Notes on X.com [61] allows users to vote for/against posts to identify misinformation and toxic speech with the wisdom of the crowd.

**Example 9.** *Suppose there are n platform users and m posts on a topic (for example, whether restrictions should be made for the COVID pandemic). Users obtain different private information from different sources, which can be generally categorized into two types, pros and cons. Each user simultaneously chooses and labels one post based on their type. The labels on each post will eventually determine the influence on the readers. A post with more supporters spreads more widely, and a post with more opponents will be announced as misinformation. Each user aims to maximize the influence of their type and plays the game strategically. What will be a stable status in such a scenario?*

The traditional Colonel Blotto game models this scenario as a centralized game, where two opposite "colonels" (for example, campaign groups) control all the users. In the Private Blotto game, on the other hand, users make their own decisions on where to deploy. This better simulates the modern social media environment where a central coordinator is generally lacking.

**Definition 3** (Private Blotto game.). *$n$ agents are competing over $m$ items. Each agent has a type (pro or con). Every agent (simultaneously) chooses exactly one item to label. The outcome of each item is determined by some outcome function. The disutility of each agent is the distance from the agent's type to each item's outcome.*

The results in the Private Blotto game [19] appear to heavily rely on the complete lack of coordination, which is also not entirely realistic. While a central coordinator is lacking, an agent can still locally coordinate with a few others. This allows (small) strategic groups and local campaigns to emerge in real-world scenarios. Moreover, these settings nearly always lack complete information and might be more faithfully modeled by agents receiving different information about various topics.

In this setting, our new solution concept of (ex-ante) Bayesian $k$-strong equilibrium seamlessly interpolates between these two extremes of complete centralization and complete decentralization. The bound $k$ can characterize how well-organized the agents are. When $k = 1$, agents are fully decentralized. A larger $k$ characterizes scenarios where agents coordinate with friends, neighborhoods, or campaigns on relevant issues. Finally, when $k$ is large enough, agents can be viewed as commanded by two opposite centralized "colonels", and the game becomes closer to the traditional Colonel Blotto game. Moreover, our definition will also naturally extend to the setting where agents have more than two sides (for example, different political factions that are more or less aligned) and the scenario where the agent's utilities are related to an underlying ground truth rather than peer partisanship.

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

## A  Properties of proper scoring rule.

For a scoring rule $PS$, we define $\tilde{PS} : \Delta_{\mathcal{S}} \times \Delta_{\mathcal{S}} \to \mathbb{R}$ such that for two distributions $\mathbf{q}_1$ and $\mathbf{q}_2$, $\tilde{PS}(\mathbf{q}_1; \mathbf{q}_2) = \mathbb{E}_{s \sim \mathbf{q}_1}[PS(s, \mathbf{q}_2)]$.

Proper scoring rules have the following properties.

**Theorem 3.** *[[31]] A scoring rule $PS$ is (strictly) proper if and only if there exists a (strictly) convex function $G : \Delta_{\mathcal{S}} \to \mathbb{R}$, such that for any $\mathbf{q}$, $G(\mathbf{q}) = \tilde{PS}(\mathbf{q}; \mathbf{q})$, and $PS(s, \mathbf{q}) = G(\mathbf{q}) + dG(\mathbf{q})(\delta_s - \mathbf{q})$, where $dG$ is a subgradient of $G$, and $\delta_s$ is the distribution that putting probability 1 on $s$.*

We use this property to prove the following Lemma.

**Lemma 3.** *Given the assumptions, the following inequalities hold. (1) $PS(h, \mathbf{q}_h) > PS(h, \mathbf{q}_\ell)$. (2) $PS(\ell, \mathbf{q}_\ell) > PS(\ell, \mathbf{q}_h)$.*

PROOF OF LEMMA 3. Suppose $G$ and $dG$ are the convex function and the subgradient satisfying Theorem 3 regarding $PS$. Then we have

$$PS(h, \mathbf{q}_h) = G(\mathbf{q}_h) + dG(\mathbf{q}_h) \cdot (\delta_h - \mathbf{q}_h).$$
$$PS(h, \mathbf{q}_\ell) = G(\mathbf{q}_\ell) + dG(\mathbf{q}_\ell) \cdot (\delta_h - \mathbf{q}_\ell).$$

$\delta_h$ is the distribution on $\mathcal{S}$ that putting probability 1 on $h$.

Since $dG$ is a subgradient of $G$, it satisfies $G(y) \geq G(x) + dG(x) \cdot (y - x)$ for any $x, y \in \Delta_{\mathcal{S}}$. Consequently, $(dG(y) - dG(x)) \cdot (y - x) \geq 0$ for any $x, y \in \Delta_{\mathcal{S}}$. Then we have

$$PS(h, \mathbf{q}_h) - PS(h, \mathbf{q}_\ell) = G(\mathbf{q}_h) - G(\mathbf{q}_\ell) + dG(\mathbf{q}_h) \cdot (\delta_h - \mathbf{q}_h) - dG(\mathbf{q}_\ell) \cdot (\delta_h - \mathbf{q}_\ell)$$
$$\geq dG(\mathbf{q}_\ell) \cdot (\mathbf{q}_h - \mathbf{q}_\ell) + dG(\mathbf{q}_h) \cdot (\delta_h - \mathbf{q}_h) - dG(\mathbf{q}_\ell) \cdot (\delta_h - \mathbf{q}_\ell)$$
$$= (dG(\mathbf{q}_h) - dG(\mathbf{q}_\ell)) \cdot (\delta_h - \mathbf{q}_h).$$

Note that $(\delta_h - \mathbf{q}_h)(h) = 1 - q(h \mid h) = q(\ell \mid h)$, and $(\delta_h - \mathbf{q}_h)(\ell) = -q(\ell \mid h)$. On the other hand, $(\mathbf{q}_h - \mathbf{q}_\ell)(h) = -(\mathbf{q}_h - \mathbf{q}_\ell)(\ell) = q(h \mid h) - q(h \mid \ell) > 0$. Therefore,

$$PS(h, \mathbf{q}_h) - PS(h, \mathbf{q}_\ell) = (dG(\mathbf{q}_h) - dG(\mathbf{q}_\ell)) \cdot (\delta_h - \mathbf{q}_h)$$
$$= \frac{q(\ell \mid h)}{q(h \mid h) - q(h \mid \ell)} \cdot (dG(\mathbf{q}_h) - dG(\mathbf{q}_\ell)) \cdot (\mathbf{q}_h - \mathbf{q}_\ell)$$
$$\geq 0,$$

With similar reasoning we have $PS(\ell, \mathbf{q}_\ell) - PS(\ell, \mathbf{q}_h) \geq 0$. Then note that since $PS$ is a strictly proper scoring rule,

$$\tilde{PS}(\mathbf{q}_h; \mathbf{q}_h) - \tilde{PS}(\mathbf{q}_h, \mathbf{q}_\ell)$$
$$= q(h \mid h) \cdot (PS(h, \mathbf{q}_h) - PS(h, \mathbf{q}_\ell)) + q(\ell \mid h) \cdot (PS(\ell, \mathbf{q}_h) - PS(\ell, \mathbf{q}_\ell))$$
$$> 0.$$

Therefore, $PS(h, \mathbf{q}_h) - PS(h, \mathbf{q}_\ell) > 0$. The proof of (2) follows a similar reasoning as (1). $\square$

## B  Equivalence with Bayesian Nash Equilibrium

In this section, we show that when $k = 1$, both ex-ante Bayesian $k$-strong equilibrium and Bayesian $k$-strong equilibrium are equivalent to the classical Bayesian Nash Equilibrium [30].

**Definition 4** (Bayesian Nash equilibrium). *A strategy profile $\Sigma = (\sigma_i)_{i \in [n]}$ is a Bayesian Nash equilibrium (BNE) if for every agent $i$, every $i$'s strategy $\sigma_i'$, and every type $s_i \in \mathcal{S}_i$, $u_i(\Sigma \mid s_i) \geq u_i((\sigma_i', \Sigma_{-i}) \mid s_i)$, where $\Sigma_{-i}$ is the strategies all other agents play in $\Sigma$.*

The following propositions shows the equivalence among solution concepts.

**Proposition 2.** *If a strategy profile $\Sigma$ is a Bayesian Nash equilibrium, then $\Sigma$ is an ex-ante Bayesian 1-strong equilibrium.*

PROOF. Suppose $\Sigma$ is a Bayesian Nash equilibrium, then for every agent $i$, every $i$'s strategy $\sigma_i'$, and every type $s_i \in \mathcal{S}_i$, $u_i(\Sigma \mid s_i) \geq u_i((\sigma_i', \Sigma_{-i}) \mid s_i)$. Then from the law of total probability, adding up all the types in $\mathcal{S}$, $u_i(\Sigma) \geq u_i((\sigma_i', \Sigma_{-i}))$. This implies that $\Sigma$ is an ex-ante Bayesian 1-strong equilibrium. $\square$

**Proposition 3.** *If a strategy profile $\Sigma$ is a Bayesian 1-strong equilibrium, then $\Sigma$ is a Bayesian Nash equilibrium.*

PROOF. Suppose $\Sigma$ is NOT a Bayesian Nash equilibrium, and for agent $i$, strategy $\sigma_i'$, and type $s_i$. $u_i(\Sigma \mid s_i) < u_i((\sigma_i', \Sigma_{-i}) \mid s_i)$. Now consider the strategy $\sigma_i''$ such that for all $s_i' \in \mathcal{S}_i$ and $s_i' \neq s_i$, $\sigma_i''(s') = \sigma_i(s')$, and $\sigma_i''(s_i) = \sigma_i'(s)$. Then we have $u_i(\Sigma \mid s_i) < u_i((\sigma_i'', \Sigma_{-i}) \mid s_i)$ and $u_i(\Sigma \mid s_i) = u_i((\sigma_i', \Sigma_{-i}) \mid s_i)$ for all $s_i' \in \mathcal{S}_i$ and $s_i' \neq s_i$. This implies that $\Sigma$ is a Bayesian 1-strong equilibrium. $\square$

Proposition 2, Proposition 3, and Proposition 1 when $k = 1$ form a cycle of equivalence.

## C  Proof of Theorem 1

The proof consists of two steps. In Step 1, we characterize $k_E$ by comparing the ex-ante expected utility of a deviator when every agent reports truthfully and when all $k$ deviators always report $h$ (and always report $\ell$, respectively). At least one of the two deviations brings a deviator higher expected utility if and only if $k > k_E$. In Step 2, we show that for any $k \le k_E$ and any deviating strategy profile $\Sigma'$, the average expected utility among all the deviators when $\Sigma'$ is played will not exceed the expected utility when every agent reports truthfully. Therefore, there exists a deviator whose expected utility will decrease after the deviation, and the deviation cannot succeed.

**Step 1: characterizing $k_E$.**

Consider a deviating group $D$ of $k$ agents. In the deviating strategy profile $\Sigma$, all the deviators always report $h$, i.e. $\sigma = (1, 1)$. (The reasoning for all deviators reporting $\ell$ will be similar.) We fix an arbitrary deviator $i \in D$ and characterize the condition of $k$ such that $u_i(\Sigma) > u_i(\Sigma^*)$. Due to the symmetricity of the strategy profile, this implies that the expected utility of every deviator is higher in deviation than in truth-telling, and the deviation is successful.

To compare the expected utilities, we divide $u_i(\Sigma)$. One part is the average expected utility from all other deviators $j \in D \setminus i$, denoted by $u_i(\Sigma \mid \text{deviator})$. The other part is the average expected utility from all truthful agents $j \in [n] \setminus D$, denoted by $u_i(\Sigma \mid \text{truthful})$.

$$u_i(\Sigma) = \frac{1}{n-1} \left( \sum_{j \in D \setminus i} \mathbb{E}[R_i(a_j)] + \sum_{j \in [n] \setminus D} \mathbb{E}[R_i(a_j)] \right)$$

$$= \frac{k-1}{n-1} \cdot u_i(\Sigma \mid \text{deviator}) + \frac{n-k}{n-1} \cdot u_i(\Sigma \mid \text{truthful}).$$

With the truthfulness of the peer prediction mechanism, $u_i(\Sigma \mid \text{truthful})$ is maximized when $i$ reports truthfully, and $i$ cannot increase his/her expected utility by deviation in this part. Therefore, agent $i$ should gain a higher expected utility in the deviation part.

Let $\Delta u_d = u_i(\Sigma^*) - u_i(\Sigma \mid \text{deviator})$, and $\Delta u_t = u_i(\Sigma^*) - u_i(\Sigma \mid \text{truthful})$. Our goal is to find the condition on $k$ such that

$$\frac{k-1}{n-1} \cdot \Delta u_d + \frac{n-k}{n-1} \cdot \Delta u_t < 0.$$

Note that when in $\Sigma$ all agents $i$ have equal expected utility. Therefore, the inequality should be strict. When $\Delta u_t > \Delta u_d$, this is equivalent to

$$k > \frac{\Delta u_t}{\Delta u_t - \Delta u_d} \cdot (n-1) + 1.$$

And when $\Delta u_t \le \Delta u_d$, the condition does not hold for any $k$, and the deviation will never succeed.

We first calculate the truthful expected utility. Note that when everyone plays the same strategy, the expected utility equals the expectation on $R_i(a_j)$.

$$u_i(\Sigma^*) = q(h) \cdot (q(h \mid h) \cdot PS(h, \mathbf{q}_h) + q(\ell \mid h) \cdot PS(\ell, \mathbf{q}_h))$$
$$+ q(\ell) \cdot (q(h \mid \ell) \cdot PS(h, \mathbf{q}_\ell) + q(\ell \mid \ell) \cdot PS(\ell, \mathbf{q}_\ell)).$$

Then we calculate $u_i(\Sigma \mid \text{truthful})$.

$$u_i(\Sigma \mid \text{truthful}) = q(h) \cdot (q(h \mid h) \cdot PS(h, \mathbf{q}_h) + q(\ell \mid h) \cdot PS(\ell, \mathbf{q}_h))$$
$$+ q(\ell) \cdot (q(h \mid \ell) \cdot PS(h, \mathbf{q}_h) + q(\ell \mid \ell) \cdot PS(\ell, \mathbf{q}_h)).$$

Therefore, the first part of the difference is

$$\Delta u_t = q(\ell) \cdot (q(h \mid \ell) \cdot (PS(h, \mathbf{q}_\ell) - PS(h, \mathbf{q}_h)) + q(\ell \mid \ell) \cdot (PS(\ell, \mathbf{q}_\ell) - PS(\ell, \mathbf{q}_h)))$$
$$= q(\ell) \cdot \mathbb{E}_{s \sim \mathbf{q}_\ell}[PS(s, \mathbf{q}_\ell) - PS(s, \mathbf{q}_h)]$$

From the property of the strict proper scoring rule, we know that $\Delta u_t > 0$.

And when $j$ is a deviator always reporting $h$, the utility of $i$ will always be $PS(h, \mathbf{q}_h)$. Therefore, the second part of the difference is

$$\Delta u_d = q(\ell) \cdot (q(h \mid \ell) \cdot (PS(h, \mathbf{q}_\ell) - PS(h, \mathbf{q}_h)) + q(\ell \mid \ell) \cdot (PS(\ell, \mathbf{q}_\ell) - PS(h, \mathbf{q}_h)))$$
$$+ q(h) \cdot q(\ell \mid h) \cdot (PS(\ell, \mathbf{q}_h) - PS(h, \mathbf{q}_h)).$$

And

$$\Delta u_t - \Delta u_d = q(\ell) \cdot q(\ell \mid \ell)(PS(h, \mathbf{q}_h) - PS(\ell, \mathbf{q}_\ell)) - q(h) \cdot q(\ell \mid h)(PS(\ell, \mathbf{q}_h) - PS(h, \mathbf{q}_h))$$
$$= q(\ell) \cdot (PS(h, \mathbf{q}_h) - PS(\ell, \mathbf{q}_h)).$$

Therefore, when $PS(h, \mathbf{q}_h) - PS(\ell, \mathbf{q}_h) > 0$, the condition for the deviation to succeed is $k > \frac{\mathbb{E}_{s \sim \mathbf{q}_\ell}[PS(s, \mathbf{q}_\ell) - PS(s, \mathbf{q}_h)]}{PS(h, \mathbf{q}_h) - PS(\ell, \mathbf{q}_h)} \cdot (n-1) + 1$. And when $PS(h, \mathbf{q}_h) - PS(\ell, \mathbf{q}_h) \le 0$, the deviation will never succeed. This is how $k_E^h$ is defined.

Similarly, for deviation where all deviators always report $\ell$, the condition for the deviation to succeed is $k > \frac{\mathbb{E}_{s \sim \mathbf{q}_h}[PS(s, \mathbf{q}_h) - PS(s, \mathbf{q}_\ell)]}{PS(\ell, \mathbf{q}_\ell) - PS(h, \mathbf{q}_\ell)} \cdot (n-1) + 1$ when $PS(\ell, \mathbf{q}_\ell) - PS(h, \mathbf{q}_\ell) > 0$, and the deviation can never succeed when $PS(\ell, \mathbf{q}_\ell) - PS(h, \mathbf{q}_\ell) \le 0$. This is how $k_E^\ell$ is defined.

Also, note that by Lemma 3, at least one of $PS(h, \mathbf{q}_h) - PS(\ell, \mathbf{q}_h) > 0$ and $PS(\ell, \mathbf{q}_\ell) - PS(h, \mathbf{q}_\ell) > 0$ holds.

Therefore, for all $k \leq k_E$, both deviations cannot succeed.

**Step 2: Equilibrium holds for $k \leq k_E$.**

We fix an arbitrary $2 \leq k \leq k_E$. (For $k = 1$, the ex-ante Bayesian 1-strong equilibrium is equivalent to BNE, which is guaranteed by the truthfulness of the peer prediction mechanism.) Let $\Sigma$ be the deviating strategy and $\sigma_i = (\beta_\ell^i, \beta_h^i)$ be the strategy agent $i$ plays in $\Sigma$. $u_i(\Sigma \mid \text{truthful})$ and $u_i(\Sigma \mid \text{deviator})$ still denote the expected utility $i$ gain from truthful agents and deviators, respectively. In this part, we consider the average on the expected utility of all agents $i \in D$. Let $\bar{\sigma} = (\bar{\beta_\ell}, \bar{\beta_h}) = \frac{1}{k} \sum_{i \in D} \sigma_i$ be the average strategy on all deviators.

For the truthful side, we have

$$u_i(\Sigma \mid \text{truthful}) = \frac{1}{n-k} \sum_{j \in [n] \setminus D} \mathbb{E}_{s_i \sim \mathbf{q}, a_i \sim \sigma_i(s_i)} \mathbb{E}_{s_j \sim \mathbf{q}_{s_i}} PS(s_j, \mathbf{q}_{a_i})$$

$$= \mathbb{E}_{s_i \sim \mathbf{q}, a_i \sim \sigma_i(s_i)} \mathbb{E}_{s_j \sim \mathbf{q}_{s_i}} PS(s_j, \mathbf{q}_{a_i})$$

Let

$$\bar{u}(\Sigma \mid \text{truthful}) = \frac{1}{k} \sum_{i \in D} u_i(\Sigma \mid \text{truthful})$$

$$= \mathbb{E}_{s \sim \mathbf{q}, a \sim \bar{\sigma}(s)} \mathbb{E}_{s_j \sim \mathbf{q}_s} PS(s_j, \mathbf{q}_a).$$

The equation comes from the fact that $\beta_h^j$ and $\beta_\ell^j$ are linear in the expected utility. Note that when everyone reports truthfully, everyone has equal expected utility, i.e. $\bar{u}(\Sigma^*) = u_i(\Sigma^*)$. Then the difference between truthful reporting and deviation is a function of $\bar{\beta_\ell}$ and $\bar{\beta_h}$.

$$\Delta u_t(\bar{\beta_\ell}, \bar{\beta_h})$$

$$= \bar{u}(\Sigma^*) - \bar{u}(\Sigma \mid \text{truthful})$$

$$= q(h) \cdot (1 - \bar{\beta_h}) \cdot (q(h \mid h) \cdot (PS(h, \mathbf{q}_h) - PS(h, \mathbf{q}_\ell)) + q(\ell \mid h) \cdot (PS(\ell, \mathbf{q}_h) - PS(\ell, \mathbf{q}_\ell)))$$

$$+ q(\ell) \cdot \bar{\beta_\ell} \cdot (q(h \mid \ell) \cdot (PS(h, \mathbf{q}_\ell) - PS(h, \mathbf{q}_h)) + q(\ell \mid \ell) \cdot (PS(\ell, \mathbf{q}_\ell) - PS(\ell, \mathbf{q}_h))).$$

$\Delta u_t(\bar{\beta_\ell}, \bar{\beta_h}) \geq 0$ and the equation holds when $\bar{\beta_h} = 1$ and $\bar{\beta_\ell} = 0$ is guaranteed by the property of the strict proper scoring rule.

For the deviator side, we have

$$u_i(\Sigma \mid \text{deviator}) = \frac{1}{k-1} \sum_{j \in D \setminus \{i\}} \mathbb{E}_{s_i \sim \mathbf{q}, a_i \sim \sigma_i(s_i)} \mathbb{E}_{s_j \sim \mathbf{q}_{s_i}, a_j \sim \sigma_j(s_j)} PS(a_j, \mathbf{q}_{a_i}).$$

And $\bar{u}(\Sigma \mid \text{deviator}) = \frac{1}{k} \sum_{i \in D} u_i(\Sigma \mid \text{deviator})$.

To better characterize $\bar{u}(\Sigma \mid \text{deviator})$, we find an upper bound parameterized only by $\bar{\sigma}$. Note that

$$u_i(\Sigma \mid \text{deviator}) = \frac{k}{k-1} \mathbb{E}_{s_i \sim \mathbf{q}, a_i \sim \sigma_i(s_i)} \mathbb{E}_{s \sim \mathbf{q}_{s_i}, a \sim \bar{\sigma}(s)} PS(a, \mathbf{q}_{a_i})$$

$$- \frac{1}{k-1} \mathbb{E}_{s_i \sim \mathbf{q}, a_i \sim \sigma_i(s_i)} \mathbb{E}_{s \sim \mathbf{q}_{s_i}, a \sim \sigma_i(s)} PS(a, \mathbf{q}_{a_i}).$$

The first term can be viewed as adding an independent deviator that plays the same strategy $\sigma_i$ as $i$ into the deviator set $D$. Then the average can be represented as

$$\bar{u}(\Sigma \mid \text{deviator}) = \frac{k}{k-1} \mathbb{E}_{s' \sim \mathbf{q}, a' \sim \bar{\sigma}(s')} \mathbb{E}_{s \sim \mathbf{q}_{s'}, a \sim \bar{\sigma}(s)} PS(a, \mathbf{q}_{a'})$$

$$- \frac{1}{(k-1)k} \sum_{i \in D} \mathbb{E}_{s_i \sim \mathbf{q}, a_i \sim \sigma_i(s_i)} \mathbb{E}_{s \sim \mathbf{q}_{s_i}, a \sim \sigma_i(s)} PS(a, \mathbf{q}_{a_i}).$$

Let $f : [0, 1]^2 \to \mathbb{R}$. For a strategy $\sigma = (\beta_\ell, \beta_h)$, let

$$f(\beta_\ell, \beta_h) = \mathbb{E}_{s' \sim \mathbf{q}, a' \sim \sigma(s')} \mathbb{E}_{s \sim \mathbf{q}_h, a \sim \sigma(s)} PS(a, \mathbf{q}_{a'}).$$

Then we can represent $\bar{u}(\Sigma \mid \text{deviator})$ in the form of $f$.

$$\bar{u}(\Sigma \mid \text{deviator}) = \frac{k}{k-1} f(\bar{\beta_\ell}, \bar{\beta_h}) - \frac{1}{(k-1)k} \sum_{i \in D} f(\beta_\ell^i, \beta_h^i).$$

**Claim 1.** $f(\beta_\ell, \beta_h)$ is convex on $[0, 1]^2$.

PROOF. Note that

$$\frac{\partial^2 f}{\partial \beta_\ell^2} = 2q(\ell) \cdot q(\ell \mid \ell) \cdot (PS(h, \mathbf{q}_h) + PS(\ell, \mathbf{q}_\ell) - PS(h, \mathbf{q}_\ell) - PS(\ell, \mathbf{q}_h)),$$

$$\frac{\partial^2 f}{\partial \beta_h^2} = 2q(h) \cdot q(h \mid h) \cdot (PS(h, \mathbf{q}_h) + PS(\ell, \mathbf{q}_\ell) - PS(h, \mathbf{q}_\ell) - PS(\ell, \mathbf{q}_h)),$$

$$\frac{\partial^2 f}{\partial \beta_\ell \partial \beta_h} = (q(\ell) \cdot q(h \mid \ell) + q(h) \cdot q(\ell \mid h))(PS(h, \mathbf{q}_h) + PS(\ell, \mathbf{q}_\ell) - PS(h, \mathbf{q}_\ell) - PS(\ell, \mathbf{q}_h)),$$

$$\frac{\partial^2 f}{\partial \beta_h \partial \beta_\ell} = (q(\ell) \cdot q(h \mid \ell) + q(h) \cdot q(\ell \mid h))(PS(h, \mathbf{q}_h) + PS(\ell, \mathbf{q}_\ell) - PS(h, \mathbf{q}_\ell) - PS(\ell, \mathbf{q}_h)).$$

Let

$$H = \begin{bmatrix} 2q(\ell) \cdot q(\ell \mid \ell) & q(\ell) \cdot q(h \mid \ell) + q(h) \cdot q(\ell \mid h) \\ q(\ell) \cdot q(h \mid \ell) + q(h) \cdot q(\ell \mid h) & 2q(h) \cdot q(h \mid h) \end{bmatrix}.$$

Then the Hermitian matrix of $f$ is

$$H(f) = (PS(h, \mathbf{q}_h) + PS(\ell, \mathbf{q}_\ell) - PS(h, \mathbf{q}_\ell) - PS(\ell, \mathbf{q}_h)) \cdot H.$$

To show the convexity of $f$, it is sufficient to show that $H(f)$ is positive semi-definite. From Lemma 3 we know that $(PS(h, \mathbf{q}_h) + PS(\ell, \mathbf{q}_\ell) - PS(h, \mathbf{q}_\ell) - PS(\ell, \mathbf{q}_h)) > 0$. Therefore, it is sufficient to show that $H$ is positive semidefinite. We leverage the following lemma.

**Lemma 4.** *[44, (7.6.12)]. A real symmetric matrix $A$ is positive semidefinite if and only if all principal minors of $A$ are non-negative.*

Therefore, it is sufficient to show that all principal minors of $H$ are non-negative.

First, $|H_{1\times1}| = 2q(\ell) \cdot q(\ell \mid \ell) > 0$, and $|H_{2\times2}| = 2q(h) \cdot q(h \mid h) > 0$. Note that $q(\ell) \cdot q(h \mid \ell) = q(h) \cdot q(\ell \mid h)$. Therefore,

$$|H| = 4q(h) \cdot q(\ell) \cdot (q(\ell \mid \ell) \cdot q(h \mid h) - q(h \mid \ell) \cdot q(\ell \mid h)) > 0.$$

Therefore, the Hessian matrix of $f$ is positive semidefinite. Consequently, $f$ is convex. □

By the convexity, $\frac{1}{k} \sum_{i \in D} f(\beta_\ell^i, \beta_h^i) \geq f(\bar{\beta}_\ell, \bar{\beta}_h)$. Therefore, $\bar{u}(\Sigma \mid \text{deviator}) \leq f(\bar{\beta}_\ell, \bar{\beta}_h)$, and the equality holds if all the agents $i \in D$ plays the same strategy.

We use $f(\bar{\beta}_\ell, \bar{\beta}_h)$ as an upper bound of $\bar{u}(\Sigma \mid \text{deviator})$. Let $\Delta u_d(\bar{\beta}_\ell, \bar{\beta}_h) = \bar{u}(\Sigma^*) - \bar{u}(\Sigma \mid \text{deviator})$, and $\Delta u'_d(\bar{\beta}_\ell, \bar{\beta}_h) = \bar{u}(\Sigma^*) - f(\bar{\beta}_\ell, \bar{\beta}_h)$. Then $\Delta u_d(\bar{\beta}_\ell, \bar{\beta}_h) \geq \Delta u'_d(\bar{\beta}_\ell, \bar{\beta}_h)$ always holds.

Now we are ready to show that truthful reporting $\Sigma^*$ is more profitable than any deviation $\Sigma$ for $k \leq k_E$.

Let $\Delta u(\bar{\beta}_\ell, \bar{\beta}_h) = \frac{k-1}{n-1} \cdot \Delta u'_d(\bar{\beta}_\ell, \bar{\beta}_h) + \frac{n-k}{n-1} \cdot \Delta u_t(\bar{\beta}_\ell, \bar{\beta}_h)$. Then it's sufficient to show that $\Delta u(\bar{\beta}_\ell, \bar{\beta}_h) \geq 0$ on any $(\bar{\beta}_\ell, \bar{\beta}_h) \in [0, 1]^2$.

First, notice that $\Delta u(\bar{\beta}_\ell, \bar{\beta}_h)$ is a concave function. This is because $\Delta u_t(\bar{\beta}_\ell, \bar{\beta}_h)$ is linear on $\bar{\beta}_\ell$ and $\bar{\beta}_h$, and $\Delta u'_d(\bar{\beta}_\ell, \bar{\beta}_h)$ is a concave function according to Claim 1. Therefore, it is sufficient to show that $\Delta u(\bar{\beta}_\ell, \bar{\beta}_h) \geq 0$ on any $(\bar{\beta}_\ell, \bar{\beta}_h) \in \{0, 1\}^2$, i.e. the corner points. Note that $\bar{\sigma} = (\bar{\beta}_\ell, \bar{\beta}_h)$ is the average strategy of all the deviators. $\bar{\beta}_\ell$ ($\bar{\beta}_h$, respectively) equals to 0 (or 1) means that for all $i \in D$, $\beta_\ell^i$ ($\beta_h^i$, respectively) equals to 0 (or 1, respectively). Therefore, in the corner points, $\Delta u'_d = \Delta u_d$.

When $\bar{\beta}_\ell = 0$ and $\bar{\beta}_h = 1$, $\Sigma = \Sigma^*$, and all the deviators also report truthfully. In this case, $\Delta u_t = \Delta u'_d = 0$ since the two strategies are the same. Therefore, $\Delta u(0, 1) = 0$.

When $\bar{\beta}_\ell = \bar{\beta}_h = 1$, all the deviators always report $h$. In Step 1 we have shown that such deviation cannot succeed for any $k \leq k_E$. Therefore, $\Delta u(1, 1) \geq 0$.

When $\bar{\beta}_\ell = \bar{\beta}_h = 0$, all the deviators always report $\ell$. In Step 1 we have shown that such deviation cannot succeed for any $k \leq k_E$. Therefore, $\Delta u(0, 0) \geq 0$.

And when $\bar{\beta}_\ell = 1$ and $\bar{\beta}_h = 0$, all the deviators always tell a lie. We follow a similar reasoning as in step 1. First, we have

$$\Delta u_t(1, 0) = q(h) \cdot (q(h \mid h) \cdot (PS(h, \mathbf{q}_h) - PS(h, \mathbf{q}_\ell)) + q(\ell \mid h) \cdot (PS(\ell, \mathbf{q}_h) - PS(\ell, \mathbf{q}_\ell)))$$
$$+ q(\ell) \cdot (q(h \mid \ell) \cdot (PS(h, \mathbf{q}_\ell) - PS(h, \mathbf{q}_h)) + q(\ell \mid \ell) \cdot (PS(\ell, \mathbf{q}_\ell) - PS(\ell, \mathbf{q}_h)))$$
$$= q(h) \cdot \mathbb{E}_{s \sim \mathbf{q}_h}[PS(s, \mathbf{q}_h) - PS(s, \mathbf{q}_\ell)] + q(\ell) \cdot \mathbb{E}_{s \sim \mathbf{q}_\ell}[PS(s, \mathbf{q}_\ell) - PS(s, \mathbf{q}_h)].$$

$\Delta u_t(1, 0) \geq 0$ is guaranteed by the property of the proper scoring rule.

And

$$\Delta u_t(1, 0) - \Delta u'_d(1, 0)$$
$$= q(h) \cdot (q(h \mid h) \cdot (PS(\ell, \mathbf{q}_\ell) - PS(h, \mathbf{q}_\ell)) + q(\ell \mid h) \cdot (PS(h, \mathbf{q}_\ell) - PS(\ell, \mathbf{q}_\ell)))$$
$$+ q(\ell) \cdot (q(h \mid \ell) \cdot (PS(\ell, \mathbf{q}_h) - PS(h, \mathbf{q}_h)) + q(\ell \mid \ell) \cdot (PS(h, \mathbf{q}_h) - PS(\ell, \mathbf{q}_h)))$$
$$= q(h) \cdot (q(h \mid h) - q(\ell \mid h)) \cdot (PS(\ell, \mathbf{q}_\ell) - PS(h, \mathbf{q}_\ell))$$
$$+ q(\ell) \cdot (q(\ell \mid \ell) - q(h \mid \ell)) \cdot (PS(h, \mathbf{q}_h) - PS(\ell, \mathbf{q}_h)).$$

If $\Delta u_t(1,0) - \Delta u'_d(1,0) \le 0$, $\Delta u(1,0) \ge 0$ for every $k$. If $\Delta u_t(1,0) - \Delta u'_d(1,0) > 0$, $\Delta u(1,0) < 0$ if and only if

$$k > \frac{q(h) \cdot \mathbb{E}_{s \sim \mathbf{q}_h}[PS(s, \mathbf{q}_h) - PS(s, \mathbf{q}_\ell)] + q(\ell) \cdot \mathbb{E}_{s \sim \mathbf{q}_\ell}[PS(s, \mathbf{q}_\ell) - PS(s, \mathbf{q}_h)]}{q(h) \cdot (q(h \mid h) - q(\ell \mid h)) \cdot (PS(\ell, \mathbf{q}_\ell) - PS(h, \mathbf{q}_\ell)) + q(\ell) \cdot (q(\ell \mid \ell) - q(h \mid \ell)) \cdot (PS(h, \mathbf{q}_h) - PS(\ell, \mathbf{q}_h))} + 1.$$

Denote this threshold as $k'$. We claim that $k' \ge k_E$, and therefore $\Delta u(1,0) \ge 0$ for all $k \le k_E$. We consider the following three cases.

(1) $PS(\ell, \mathbf{q}_\ell) - PS(h, \mathbf{q}_\ell) > 0$ and $PS(h, \mathbf{q}_h) - PS(\ell, \mathbf{q}_h) > 0$. In this case, $k'$ is between $k_E^h$ and $k_E^\ell$, and cannot be smaller than $k_E$.

(2) $PS(\ell, \mathbf{q}_\ell) - PS(h, \mathbf{q}_\ell) \le 0$ but $PS(h, \mathbf{q}_h) - PS(\ell, \mathbf{q}_h) > 0$. If $q(h \mid h) \ge q(\ell \mid h)$, the first term in the denominator is non-positive, and it's not hard to verify that $k' > k_E$. If $q(h \mid h) < q(\ell \mid h)$, note that we have $PS(h, \mathbf{q}_h) > PS(h, \mathbf{q}_\ell) \ge PS(\ell, \mathbf{q}_\ell) > PS(\ell, \mathbf{q}_h)$ according to Lemma 3. Therefore, the denominator is no more than $(q(\ell) \cdot q(\ell \mid \ell) - q(h) \cdot q(h \mid h)) \cdot PS(h, \mathbf{q}_h) - PS(\ell, \mathbf{q}_h))$, which is smaller than $q(\ell)$ times the denominator in the $k_E^h$. On the other hand, the nominator in $k'$ is larger than $q(\ell)$ times the nominator in $k_E$. Therefore, $k' \ge k_E$.

(3) $PS(\ell, \mathbf{q}_\ell) - PS(h, \mathbf{q}_\ell) > 0$ but $PS(h, \mathbf{q}_h) - PS(\ell, \mathbf{q}_h) \le 0$. This case follows the second case due to symmetricity.

Therefore, $\Delta u(1,0) \ge 0$ for all $k \le k_E$.

Therefore, we prove that for any $k \le k_E$ and any strategy profile, the average (ex-ante) expected utility of the deviators will not exceed the expected utility when all the agents report truthfully. Therefore, one of the following cases occurs.

(1) There exists an agent $i$ such that $i$'s expected utility in deviation is strictly lower than in $\Sigma^*$. Therefore, the second condition of the deviating group is violated.

(2) All the agents have exactly the same expected utility in deviation and $\Sigma^*$. The third condition of a deviating group to have an agent strictly better off is violated.

Therefore, any deviation cannot succeed, and truth-telling $\Sigma^*$ is an ex-ante Bayesian $k_E$-strong equilibrium.

## D  Proof of Theorem 2

The steps of the proof resemble the steps of the proof of Theorem 1, yet the techniques are different. In Step 1, we determine $k_B$ by comparing the interim expected utility of a deviator conditioned on his/her signal being $h$ and $\ell$ respectively when every agent reports truthfully and when all $k$ deviators always report $h$ (and always report $\ell$, respectively). In Step 2, we show that for any $k \le k_E$ and any deviating strategy profile $\bar{\Sigma}$ where all the deviators play the same strategy $\bar{\sigma}$, the average expected utility among all the deviators when $\bar{\Sigma}$ is played will not exceed the expected utility when every agent reports truthfully. In Step 3, we show that for sufficiently large $n$, any $k \le k_B$, and any deviating strategy profile $\Sigma$, there exists a deviator whose expected utility is strictly smaller than the expected utility when every agent reports truthfully.

Let $\overline{\Delta PS}$ be the largest difference in the positional scoring rule. Let $\Delta h = PS(h, \mathbf{q}_h) - PS(h, \mathbf{q}_\ell)$, and $\Delta \ell = PS(\ell, \mathbf{q}_\ell) - q(\ell, \mathbf{q}_h)$. From Lemma 3, we have $\Delta h > 0$ and $\Delta \ell > 0$. We first explicitly give the lower bound of $n$:

(1) $\bar{b}_h = \frac{4\overline{\Delta PS} \cdot (\Delta h + \Delta \ell + PS(\ell, \mathbf{q}_\ell) - PS(h, \mathbf{q}_\ell))}{(n-1) \cdot (\Delta h + \Delta \ell) \cdot \mathbb{E}_{s \sim \mathbf{q}_h}[PS(s, \mathbf{q}_h) - q(s, \mathbf{q}_\ell)]} < \frac{1}{4}$,

(2) If $PS(h, \mathbf{q}_h) > PS(\ell, \mathbf{q}_h)$, then $\bar{b}_h \le \frac{PS(h, \mathbf{q}_h) - PS(\ell, \mathbf{q}_h)}{\Delta h + \Delta \ell}$,

(3) $\bar{b}_h \le \frac{\mathbb{E}_{s \sim \mathbf{q}_h}[PS(s, \mathbf{q}_h) - q(s, \mathbf{q}_\ell)]}{q(h \mid h) \cdot (\Delta h + \Delta \ell)}$,

(4) $\bar{b}_l = \frac{4\overline{\Delta PS} \cdot (\Delta h + \Delta \ell + PS(h, \mathbf{q}_h) - PS(\ell, \mathbf{q}_h))}{(n-1) \cdot (\Delta h + \Delta \ell) \cdot \mathbb{E}_{s \sim \mathbf{q}_\ell}[PS(s, \mathbf{q}_\ell) - q(s, \mathbf{q}_h)]} < \frac{1}{4}$,

(5) If $PS(\ell, \mathbf{q}_\ell) > PS(h, \mathbf{q}_\ell)$, then $\bar{b}_h \le \frac{PS(\ell, \mathbf{q}_\ell) - PS(h, \mathbf{q}_\ell)}{\Delta h + \Delta \ell}$,

(6) $\bar{b}_h \le \frac{\mathbb{E}_{s \sim \mathbf{q}_\ell}[PS(s, \mathbf{q}_\ell) - q(s, \mathbf{q}_h)]}{q(\ell \mid \ell) \cdot (\Delta h + \Delta \ell)}$.

**Step 1: characterizing $k_B$.**

Consider a deviating group $D$ of $k$ agents. In the deviating strategy profile $\Sigma$, all the deviators always report $h$, i.e. $\sigma = (1, 1)$. We fix an arbitrary deviator $i \in D$. In the interim setting, the condition for the deviation is successful is $u_i(\Sigma \mid h) \ge u_i(\Sigma^* \mid h)$ and $u_i(\Sigma \mid \ell) \ge u_i(\Sigma^* \mid \ell)$ hold, and at least one of the inequality is strict.

Similar to the ex-ante's proof, let $u_i(\Sigma \mid s_i, \text{deviator})$ $u_i(\Sigma \mid s_i, \text{truthful})$ be the average expected utility from all other deviators (truthful agents, respectively) conditioned on $i$'s signal being $s_i$. And let $\Delta u_{d \mid s_i} = u_i(\Sigma^* \mid s_i) - u_i(\Sigma \mid s_i, \text{deviator})$ and $\Delta u_{t \mid s_i} = u_i(\Sigma^* \mid s_i) - u_i(\Sigma \mid s_i, \text{truthful})$.

For expected utility on $\Sigma^*$, we have

$$u_i(\Sigma^* \mid h) = q(h \mid h) \cdot PS(h, \mathbf{q}_h) + q(\ell \mid h) \cdot PS(\ell, \mathbf{q}_h),$$
$$u_i(\Sigma^* \mid \ell) = q(h \mid \ell) \cdot PS(h, \mathbf{q}_\ell) + q(\ell \mid \ell) \cdot PS(\ell, \mathbf{q}_\ell).$$

And for expected utility of $\Sigma$, we have

$$u_i(\Sigma \mid h, \text{truthful}) = q(h \mid h) \cdot PS(h, \mathbf{q}_h) + q(\ell \mid h) \cdot PS(\ell, \mathbf{q}_h)$$
$$u_i(\Sigma \mid \ell, \text{truthful}) = q(h \mid \ell) \cdot PS(h, \mathbf{q}_h) + q(\ell \mid \ell) \cdot PS(\ell, \mathbf{q}_h).$$

Therefore, $u_i(\Sigma^* \mid h) = u_i(\Sigma \mid h, \text{truthful})$, and $\Delta u_{t \mid h} = 0$. Also for the $\ell$ side, $\Delta u_{t \mid \ell} = \mathbb{E}_{s \sim \mathbf{q}_\ell}[PS(s, \mathbf{q}_\ell) - PS(s, \mathbf{q}_h)] > 0$.

For the deviator's part, we have $u_i(\Sigma \mid h, \text{deviator}) = u_i(\Sigma \mid \ell, \text{deviator}) = PS(h, \mathbf{q}_h)$.

Then we have

$$\Delta u_{t|h} - \Delta u_{d|h} = q(\ell \mid h) \cdot (PS(h, \mathbf{q}_h) - PS(\ell, \mathbf{q}_h)).$$

$$\Delta u_{t|\ell} - \Delta u_{d|\ell} = q(\ell \mid \ell) \cdot (PS(h, \mathbf{q}_h) - PS(\ell, \mathbf{q}_h)).$$

When $PS(h, \mathbf{q}_h) \leq PS(\ell, \mathbf{q}_h)$, neither agents with private signal $h$ nor those with $\ell$ can get strictly positive expected utility via deviation, and the deviation cannot succeed. When $PS(h, \mathbf{q}_h) > PS(\ell, \mathbf{q}_h)$, for the $h$ side, the condition for deviation to achieve non-negative expected utility (not strictly positive!) is

$$k \geq \frac{\Delta u_{t|h}}{\Delta u_{t|h} - \Delta u_{d|h}} \cdot (n-1) + 1 = 1,$$

and for the $\ell$ side, the condition is

$$k \geq \frac{\Delta u_{t|\ell}}{\Delta u_{t|\ell} - \Delta u_{d|\ell}} \cdot (n-1) + 1 = \frac{\mathbb{E}_{s \sim \mathbf{q}_\ell}[PS(s, \mathbf{q}_\ell) - PS(s, \mathbf{q}_h)]}{q(\ell \mid \ell) \cdot (PS(h, \mathbf{q}_h) - PS(\ell, \mathbf{q}_h))} \cdot (n-1) + 1 > 1.$$

When $k \geq \frac{\mathbb{E}_{s \sim \mathbf{q}_\ell}[PS(s, \mathbf{q}_\ell) - PS(s, \mathbf{q}_h)]}{q(\ell \mid \ell) \cdot (PS(h, \mathbf{q}_h) - PS(\ell, \mathbf{q}_h))} \cdot (n-1) + 1$, deviators with signal $h$ get strictly higher expected utility, and deviators with $\ell$ get non-decreasing expected utility. Therefore, the deviation will succeed. This is how $k_B^h$ is defined. $k_B^\ell$ is defined similarly by consider strategy $(0, 0)$. Therefore, for all $k \leq k_B$, both deviations of always reporting $h$ and always reporting $\ell$ cannot succeed.

**Step 2: Symmetric deviaton cannot succeed for $k \leq k_B$.**

We first introduce a function that will be widely applied in Step 2 and 3. Let $f^h(\beta'_h, (\beta_\ell, \beta_h)) : \mathbb{R} \times \mathbb{R}^2 \to \mathbb{R}$ be the expected reward of an agent $i$ who has signal $h$ and will report $h$ with probability $\beta'_h$, given that $i$'s peer $j$ plays the strategy $\sigma = (\beta_\ell, \beta_h)$.

$$f^h(\beta'_h, (\beta_\ell, \beta_h)) = \mathbb{E}_{a_i \sim \beta'_h} \mathbb{E}_{s_j \sim \mathbf{q}_h, a_j \sim (\beta_\ell, \beta_h)} PS(a_j, \mathbf{q}_{a_i})$$

$$= \beta'_h((q(h \mid h) \cdot \beta_h + q(\ell \mid h) \cdot \beta_\ell) \cdot PS(h, \mathbf{q}_h)$$

$$+ (1 - q(h \mid h) \cdot \beta_h - q(\ell \mid h) \cdot \beta_\ell) \cdot PS(\ell, \mathbf{q}_h))$$

$$+ (1 - \beta'_h) \cdot ((q(h \mid h) \cdot \beta_h + q(\ell \mid h) \cdot \beta_\ell) \cdot PS(h, \mathbf{q}_\ell)$$

$$+ (1 - q(h \mid h) \cdot \beta_h - q(\ell \mid h) \cdot \beta_\ell) \cdot PS(\ell, \mathbf{q}_\ell)).$$

Similarly, we define $f^\ell$.

$$f^\ell(\beta'_\ell, (\beta_\ell, \beta_h)) = \mathbb{E}_{a_i \sim \beta'_\ell} \mathbb{E}_{s_j \sim \mathbf{q}_\ell, a_j \sim (\beta_\ell, \beta_h)} PS(a_j, \mathbf{q}_{a_i})$$

$$= \beta'_\ell((q(h \mid \ell) \cdot \beta_h + q(\ell \mid \ell) \cdot \beta_\ell) \cdot PS(h, \mathbf{q}_h)$$

$$+ (1 - q(h \mid \ell) \cdot \beta_h - q(\ell \mid \ell) \cdot \beta_\ell) \cdot PS(\ell, \mathbf{q}_h))$$

$$+ (1 - \beta'_\ell) \cdot ((q(h \mid \ell) \cdot \beta_h + q(\ell \mid \ell) \cdot \beta_\ell) \cdot PS(h, \mathbf{q}_\ell)$$

$$+ (1 - q(h \mid \ell) \cdot \beta_h - q(\ell \mid \ell) \cdot \beta_\ell) \cdot PS(\ell, \mathbf{q}_\ell)).$$

Moreover, let $g^h(\beta_\ell, \beta_h) = f^h(\beta_h, (\beta_\ell, \beta_h))$, and $g^\ell(\beta_\ell, \beta_h) = f^\ell(\beta_\ell, (\beta_\ell, \beta_h))$. $g^h$ and $g^\ell$ cover the special case where $i$ and $j$ play the same strategy and will be largely applied in Step 2.

**Claim 2.** *We claim that $f^h$ and $g^h$ has the following properties.*

(1) $\frac{\partial^2 g^h}{\partial \beta_h^2} = 2q(h \mid h) \cdot (PS(h, \mathbf{q}_h) - PS(h, \mathbf{q}_\ell) - PS(\ell, \mathbf{q}_h) + PS(\ell, \mathbf{q}_\ell)) > 0.$

(2) $\frac{\partial^2 g^h}{\partial \beta_\ell^2} = 0.$

(3) *Let $\alpha_h = \frac{q(h|h)}{q(\ell|h)}$, and let $b_h \geq 0$ be a constant. When fixing $\beta_\ell = \alpha_h \cdot (b_h - \beta_h)$, then*

$$\frac{\partial g^h(\alpha_h(b_h - \beta_h), \beta_h)}{\partial \beta_h} = \frac{\partial f^h(\beta'_h, (\alpha_h(b_h - \beta_h), \beta_h))}{\partial \beta'_h}$$

$$= b_h \cdot q(h \mid h)(PS(h, \mathbf{q}_h) - PS(h, \mathbf{q}_\ell)) + (1 - b_h \cdot q(h \mid h))(PS(\ell, \mathbf{q}_h) - PS(\ell, \mathbf{q}_\ell)).$$

*Specifically, when $b_h = 1$, $\frac{\partial f^h(\beta'_h, (\alpha_h(b_h - \beta_h), \beta_h))}{\partial \beta'_h} = \mathbb{E}_{s \sim \mathbf{q}_h}[PS(s, \mathbf{q}_h) - PS(s, \mathbf{q}_\ell)] > 0.$*

**Claim 3.** *We claim that $f^\ell$ and $g^\ell$ has the following properties.*

(1) $\frac{\partial^2 g^\ell}{\partial \beta_\ell^2} = 2q(\ell \mid \ell) \cdot (PS(h, \mathbf{q}_h) - PS(h, \mathbf{q}_\ell) - PS(\ell, \mathbf{q}_h) + PS(\ell, \mathbf{q}_\ell)) > 0.$

(2) $\frac{\partial^2 g^\ell}{\partial \beta_h^2} = 0.$

(3) Let $\alpha_\ell = \frac{q(\ell|\ell)}{q(h|\ell)}$, and let $b_\ell \geq 0$ be a constant. When fixing $\beta_h = (b_\ell - \alpha_\ell \cdot \beta_\ell)$, then

$$\frac{\partial g^\ell(\beta_\ell, (b_\ell - \alpha_\ell \cdot \beta_\ell))}{\partial \beta_\ell} = \frac{\partial f^\ell(\beta'_\ell, (\beta_\ell, (b_\ell - \alpha_\ell \cdot \beta_\ell)))}{\partial \beta'_\ell}$$

$$= b_\ell \cdot q(h \mid \ell)(PS(h, \mathbf{q}_h) - PS(h, \mathbf{q}_\ell)) + (1 - b_\ell \cdot q(h \mid \ell))(PS(\ell, \mathbf{q}_h) - PS(\ell, \mathbf{q}_\ell)).$$

Specifically, when $b_\ell = 1$, $\frac{\partial f^\ell(\beta'_\ell, (\beta_\ell, (b_\ell - \alpha_\ell \cdot \beta_\ell)))}{\partial \beta'_\ell} = \mathbb{E}_{s \sim \mathbf{q}_\ell}[PS(s, \mathbf{q}_h) - PS(s, \mathbf{q}_\ell)] < 0.$

Now we start to characterize the deviation. We fix an arbitrary $k$. Let $\bar{\sigma} = (\bar{\beta}_\ell, \bar{\beta}_h)$ be the strategy on all deviators. Since all the deviators play the same strategy, they receive the same expected utility conditioned on the same signal.

The expected utility of deviator with private signal $h$ conditioned on his/her peer $j$ is a truthful agent is $u(\bar{\beta}_\ell, \bar{\beta}_h \mid h, \text{truthful}) = f^h(\bar{\beta}_h, (0, 1))$, and that conditioned on $j$ is also a deviator is $u(\bar{\beta}_\ell, \bar{\beta}_h \mid h, \text{deviator}) = f^h(\bar{\beta}_h, (\bar{\beta}_\ell, \bar{\beta}_h)) = g^h(\bar{\beta}_\ell, \bar{\beta}_h)$. Similarly, for the $\ell$ side we have $u(\bar{\beta}_\ell, \bar{\beta}_h \mid \ell, \text{truthful}) = f^\ell(\bar{\beta}_\ell, (0, 1))$ and $u(\bar{\beta}_\ell, \bar{\beta}_h \mid \ell, \text{deviator}) = f^\ell(\bar{\beta}_\ell, (\bar{\beta}_\ell, \bar{\beta}_h)) = g^\ell(\bar{\beta}_\ell, \bar{\beta}_h)$. Therefore, the expected reward of deviation can be represented by the following function.

$$u(\bar{\beta}_\ell, \bar{\beta}_h \mid h) = \frac{n - k}{n - 1} \cdot u(\bar{\beta}_\ell, \bar{\beta}_h \mid h, \text{truthful}) + \frac{k - 1}{n - 1} \cdot u(\bar{\beta}_\ell, \bar{\beta}_h \mid h, \text{deviator}),$$

$$u(\bar{\beta}_\ell, \bar{\beta}_h \mid \ell) = \frac{n - k}{n - 1} \cdot u(\bar{\beta}_\ell, \bar{\beta}_h \mid \ell, \text{truthful}) + \frac{k - 1}{n - 1} \cdot u(\bar{\beta}_\ell, \bar{\beta}_h \mid \ell, \text{deviator}).$$

Now we show that for any $\bar{\sigma} = (\bar{\beta}_\ell, \bar{\beta}_h) \in [0, 1]^2$, either $u(\bar{\beta}_\ell, \bar{\beta}_h \mid h) \leq u(\Sigma^* \mid h)$ or $u(\bar{\beta}_\ell, \bar{\beta}_h \mid \ell) \leq u(\Sigma^* \mid \ell)$.

**Lemma 1.** For any $(\bar{\beta}_\ell, \bar{\beta}_h) \in \mathbb{R}^2$ satisfying (1) $\bar{\beta}_\ell \geq 0$, (2) $\bar{\beta}_h \geq 0$, and (3) $\bar{\beta}_h + \frac{q(\ell|h)}{q(h|h)} \cdot \bar{\beta}_\ell \leq 1$, it always holds that $u(\bar{\beta}_\ell, \bar{\beta}_h \mid h) \leq u(\Sigma^* \mid h)$, and the equality holds only when $\bar{\beta}_\ell = 0$ and $\bar{\beta}_h = 1$.

PROOF OF LEMMA 1. The proof proceeds in three steps.

First, we show that $u(0, \bar{\beta}_h \mid h) \leq u(\Sigma^* \mid h)$ for any $\bar{\beta}_h$. This holds for the following three reasons. First, $u(0, 1 \mid h) = u(\Sigma^* \mid h)$, as in this case all the deviators report truthfully and no deviation happens. Second, $u(0, 0 \mid h) < u(\Sigma^* \mid h)$ is guaranteed by $k \leq k_B$. Finally, since $u(\bar{\beta}_\ell, \bar{\beta}_h \mid h, \text{truthful})$ is linear and $u(\bar{\beta}_\ell, \bar{\beta}_h \mid h, \text{deviator})$ is strictly convex on $\bar{\beta}_h$, $u(0, \bar{\beta}_h \mid h)$ is also convex on $\bar{\beta}_h$. The convexity bound the expected utility for every $0 < \bar{\beta}_h < 1$.

Second, we show that $u(\bar{\beta}_\ell, \bar{\beta}_h \mid h) \leq u(\Sigma^* \mid h)$ when $\bar{\beta}_\ell = \frac{q(h|h)}{q(\ell|h)}(1 - \bar{\beta}_h)$ for any $\bar{\beta}_h \in [0, 1]$. This is because the derivative

$$\frac{\partial u(\frac{q(h|h)}{q(\ell|h)}(1 - \bar{\beta}_h), \bar{\beta}_h \mid h)}{\partial \bar{\beta}_h} = \frac{n - k}{n - 1} \cdot \frac{\partial f^h(\bar{\beta}_h, (0, 1))}{\partial \bar{\beta}_h} + \frac{k - 1}{n - 1} \cdot \frac{\partial g^h(\frac{q(h|h)}{q(\ell|h)}(1 - \bar{\beta}_h), \bar{\beta}_h)}{\partial \bar{\beta}_h}$$

$$= \mathbb{E}_{s \sim \mathbf{q}_h}[PS(s, \mathbf{q}_h) - PS(s, \mathbf{q}_\ell)]$$

$$> 0.$$

Therefore, For any $(\bar{\beta}_\ell, \bar{\beta}_h)$ with $\bar{\beta}_h < 1$ and $\bar{\beta}_\ell = \frac{q(h|h)}{q(\ell|h)}(1 - \bar{\beta}_h)$, $u(\bar{\beta}_\ell, \bar{\beta}_h \mid h) < u(0, 1 \mid h) = u(\Sigma^* \mid h)$.

Finally, we extend the result to any $(\bar{\beta}_\ell, \bar{\beta}_h)$ in the area. For any $\bar{\beta}_h \in [0, 1)$, we have shown that $u(0, \bar{\beta}_h \mid h) \leq u(\Sigma^* \mid h)$ and $u(\frac{q(h|h)}{q(\ell|h)}(1 - \bar{\beta}_h), \bar{\beta}_h \mid h) < u(\Sigma^* \mid h)$. Then by the linearity of $u(\bar{\beta}_\ell, \bar{\beta}_h \mid h)$ on $\bar{\beta}_\ell$, $u(\bar{\beta}_\ell, \bar{\beta}_h \mid h) < u(\Sigma^* \mid h)$ for any $\bar{\beta}_\ell \in (0, \frac{q(h|h)}{q(\ell|h)}(1 - \bar{\beta}_h))$, which finishes the proof. □

Similarly, for $\ell$ side, we have

**Lemma 2.** For any $(\bar{\beta}_\ell, \bar{\beta}_h) \in \mathbb{R}^2$ satisfying (1) $\bar{\beta}_\ell \leq 1$, (2) $\bar{\beta}_h \leq 1$, and (3) $\bar{\beta}_h + \frac{q(\ell|\ell)}{q(h|\ell)} \cdot \bar{\beta}_\ell \geq 1$, it always holds that $u(\bar{\beta}_\ell, \bar{\beta}_h \mid \ell) \leq u(\Sigma^* \mid \ell)$, and the equality holds only when $\bar{\beta}_\ell = 0$ and $\bar{\beta}_h = 1$.

PROOF OF LEMMA 2. The proof follows the proof of Lemma 1. First, $u(\bar{\beta}_\ell, 1 \mid \ell) < u(\Sigma^* \mid \ell)$ for any $\bar{\beta}_\ell \in [0, 1]$ due to the convexity of $u(\bar{\beta}_\ell, \bar{\beta}_h \mid \ell)$ on $\bar{\beta}_\ell$. Secondly, $u(\bar{\beta}_\ell, \bar{\beta}_h \mid \ell) < u(\Sigma^* \mid \ell)$ when $\bar{\beta}_h + \frac{q(\ell|\ell)}{q(h|\ell)} \cdot \bar{\beta}_\ell = 1$ and $\bar{\beta}_\ell > 0$, the derivative

$$\frac{\partial u(\bar{\beta}_\ell, 1 - \frac{q(\ell|\ell)}{q(h|\ell)} \cdot \bar{\beta}_\ell \mid \ell)}{\partial \bar{\beta}_\ell} = \frac{n - k}{n - 1} \cdot \frac{\partial f^\ell(\bar{\beta}_\ell, (0, 1))}{\partial \bar{\beta}_\ell} + \frac{k - 1}{n - 1} \cdot \frac{\partial g^\ell(\bar{\beta}_\ell, 1 - \frac{q(\ell|\ell)}{q(h|\ell)} \cdot \bar{\beta}_\ell)}{\partial \bar{\beta}_\ell}$$

$$= \mathbb{E}_{s \sim \mathbf{q}_\ell}[PS(s, \mathbf{q}_h) - PS(s, \mathbf{q}_\ell)]$$

$$< 0.$$

Therefore, for any $\bar{\beta}_\ell \geq 0$ on the line, the expected reward does not exceed $u(\Sigma^* \mid \ell)$. Finally, for any other $(\bar{\beta}_\ell, \bar{\beta}_h)$ in the area, we apply the linearity of $u(\bar{\beta}_\ell, \bar{\beta}_h \mid \ell)$ on $\bar{\beta}_h$. □

Note that for any pair of $(\bar\beta_\ell, \bar\beta_h) \in [0,1]^2$, at least one of $\bar\beta_h + \frac{q(\ell|h)}{q(h|h)} \cdot \bar\beta_\ell \le 1$ and $\bar\beta_h + \frac{q(\ell|\ell)}{q(h|\ell)} \cdot \bar\beta_\ell \ge 1$ holds. This comes from $q(h \mid h) > q(h \mid \ell)$ and $q(\ell \mid \ell) > q(\ell \mid h)$. Therefore, the two triangle areas cover the whole $[0,1]^2$, and we can apply either Lemma 1 or 2 to show that the deviation cannot succeed for any $\bar\Sigma$.

**Step 3: General deviation cannot succeed for $k \le k_B$.**

Let $\overline{\Delta PS}$ be the largest difference in the positional scoring rule. Let $\Delta h = PS(h, \mathbf{q}_h) - PS(h, \mathbf{q}_\ell)$, and $\Delta\ell = PS(\ell, \mathbf{q}_\ell) - q(\ell, \mathbf{q}_h)$. From Lemma 3, we have $\Delta h > 0$ and $\Delta\ell > 0$. Then we explicitly give the lower bound of $n$:

(1) $\bar b_h = \frac{4\overline{\Delta PS} \cdot (\Delta h + \Delta\ell + PS(\ell, \mathbf{q}_\ell) - PS(h, \mathbf{q}_\ell))}{(n-1) \cdot (\Delta h + \Delta\ell) \cdot \mathbb{E}_{s \sim \mathbf{q}_h}[PS(s, \mathbf{q}_h) - q(s, \mathbf{q}_\ell)]} < \frac{1}{4}$,

(2) If $PS(h, \mathbf{q}_h) > PS(\ell, \mathbf{q}_h)$, then $\bar b_h \le \frac{PS(h, \mathbf{q}_h) - PS(\ell, \mathbf{q}_h)}{\Delta h + \Delta\ell}$,

(3) $\bar b_h \le \frac{\mathbb{E}_{s \sim \mathbf{q}_h}[PS(s, \mathbf{q}_h) - q(s, \mathbf{q}_\ell)]}{q(h|h) \cdot (\Delta h + \Delta\ell)}$,

(4) $\bar b_l = \frac{4\overline{\Delta PS} \cdot (\Delta h + \Delta\ell + PS(h, \mathbf{q}_\ell) - PS(\ell, \mathbf{q}_h))}{(n-1) \cdot (\Delta h + \Delta\ell) \cdot \mathbb{E}_{s \sim \mathbf{q}_\ell}[PS(s, \mathbf{q}_\ell) - q(s, \mathbf{q}_h)]} < \frac{1}{4}$,

(5) If $PS(\ell, \mathbf{q}_\ell) > PS(h, \mathbf{q}_\ell)$, then $\bar b_h \le \frac{PS(\ell, \mathbf{q}_\ell) - PS(h, \mathbf{q}_\ell)}{\Delta h + \Delta\ell}$,

(6) $\bar b_h \le \frac{\mathbb{E}_{s \sim \mathbf{q}_\ell}[PS(s, \mathbf{q}_\ell) - q(s, \mathbf{q}_h)]}{q(\ell|\ell) \cdot (\Delta h + \Delta\ell)}$.

Step 3 proceeds as follows. First, when deviators play asymmetrically, we compare the worst expected utility among the agent with the expected utility when all deviators play their "average" strategy $\bar\sigma = \sum_{i \in D} \frac{1}{k} \sigma_i$. We show that the reward of the worst agent cannot be better than the reward of the average strategy by $O(\frac{\overline{\Delta PS}}{n-1})$. Then we show that the reward of the average strategy is no less than the truthful reward by $\Theta(\frac{\overline{\Delta PS}}{n-1})$ only when $\bar\sigma$ is close to $(0, 1)$ or $(0, 0)$. Finally, we deal with the corner cases and show that in this case, the worst agent cannot be better than the truthful reward. We will give the proof on the $h$ side (or specifically, conditioned on an agent having private signal $h$). The $\ell$ side follows similar reasoning.

Now let $\bar\sigma = (\bar\beta_\ell, \bar\beta_h)$ be the *average strategy* of all the deviators, i.e. $\bar\sigma = \frac{1}{k}\sum_{j \in D} \sigma_j$. And $\sigma_i = (\beta_\ell^i, \beta_h^i)$ be the strategy of a deviator $i \in D$. We represent the expected utility of $i$ with $f^h$, $f^\ell$, $g^h$, and $g^\ell$.

The expected utility of $i$ conditioned on his/her peer $j$ is a truthful agent is $u(\beta_\ell^i, \beta_h^i \mid h, \text{truthful}) = f^h(\beta_h^i, (0, 1))$. and that conditioned on $j$ is also a deviator is

$$u(\beta_\ell^i, \beta_h^i \mid h, \text{deviator}) = \frac{1}{k-1} \sum_{j \in D, j \ne i} f(\beta_h^i, (\beta_\ell^j, \beta_h^j)).$$

An important observation is that for a fixed $\beta_h'$, $f^h(\beta_h', (\beta_\ell, \beta_h))$ is linear on $\beta_\ell$ and $\beta_h$. Therefore,

$$u(\beta_\ell^i, \beta_h^i \mid h, \text{deviator}) = \frac{k}{k-1} f(\beta_h^i, (\bar\beta_\ell, \bar\beta_h)) - \frac{1}{k-1} f(\beta_h^i, (\beta_\ell^i, \beta_h^i))$$

$$= f(\beta_h^i, (\bar\beta_\ell, \bar\beta_h)) + \frac{1}{k-1}(f(\beta_h^i, (\bar\beta_\ell, \bar\beta_h)) - f(\beta_h^i, (\beta_\ell^i, \beta_h^i))).$$

Adding two parts together, we have

$$u(\beta_\ell^i, \beta_h^i \mid h) = \frac{n-k}{n-1} \cdot u(\beta_\ell^i, \beta_h^i \mid h, \text{truthful}) + \frac{k-1}{n-1} \cdot u(\beta_\ell^i, \beta_h^i \mid h, \text{deviator}).$$

Then we consider the difference of agent $i$'s utility between when all the deviators play the average strategy $\bar\sigma$ and when the deviators play differently with $i$ playing $\sigma_i$.

$$u(\beta_\ell^i, \beta_h^i \mid h) - u(\bar\beta_\ell, \bar\beta_h \mid h) = \frac{n-k}{n-1} \cdot (f^h(\beta_h^i, (0, 1)) - f^h(\bar\beta_h, (0, 1)))$$

$$+ \frac{k-1}{n-1}(f(\beta_h^i, (\bar\beta_\ell, \bar\beta_h)) - f(\bar\beta_h, (\bar\beta_\ell, \bar\beta_h)))$$

$$+ \frac{k-1}{n-1} \cdot \frac{1}{k-1}(f(\beta_h^i, (\bar\beta_\ell, \bar\beta_h)) - f(\beta_h^i, (\beta_\ell^i, \beta_h^i)))$$

$$= \frac{n-k}{n-1} \cdot f^h(\beta_h^i - \bar\beta_h, (0, 1)) + \frac{k-1}{n-1} f(\beta_h^i - \bar\beta_h, (\bar\beta_\ell, \bar\beta_h))$$

$$+ \frac{1}{n-1}(f(\beta_h^i, (\bar\beta_\ell, \bar\beta_h)) - f(\beta_h^i, (\beta_\ell^i, \beta_h^i))).$$

The second equality comes from the fact that $f^h(\beta_h', (\beta_\ell, \beta_h))$ is linear on $\beta_h'$ for any fixed $(\beta_\ell, \beta_h)$. Given a fixed $\bar\sigma = (\bar\beta_\ell, \bar\beta_h)$, the term

$$\frac{n-k}{n-1} \cdot f^h(\beta_h^i - \bar\beta_h, (0, 1)) + \frac{k-1}{n-1} f(\beta_h^i - \bar\beta_h, (\bar\beta_\ell, \bar\beta_h))$$

equals to 0 when $\beta_h^i = \bar{\beta}_h$ and is linear on $\beta_h^i$. Therefore, in at least on of $\beta_h^i \leq \bar{\beta}_h$ or $\beta_\ell^i \leq \bar{\beta}_\ell$, the term will be no larger than zero.

On the other hand, the third term $\frac{1}{n-1}(f(\beta_h^i, (\bar{\beta}_\ell, \bar{\beta}_h)) - f(\beta_h^i, (\beta_\ell^i, \beta_h^i))) \leq \frac{\overline{\Delta PS}}{n-1}$. Therefore, there exists a deviator $i$ such that $u(\beta_\ell^i, \beta_h^i \mid h) - u(\bar{\beta}_\ell, \bar{\beta}_h \mid h) \leq \frac{\overline{\Delta PS}}{n-1}$.

Then we show that for sufficiently large $n$, for all $(\bar{\beta}_\ell, \bar{\beta}_h)$ not close to $(0, 1)$ or $(0, 0)$, $u(\Sigma^* \mid h) - u(\bar{\beta}_\ell, \bar{\beta}_h \mid h) > \frac{\overline{\Delta PS}}{n-1}$.

**Lemma 5.** *Let* $\Delta u(\bar{\beta}_\ell, \bar{\beta}_h \mid h) = u(\Sigma^* \mid h) - u(\bar{\beta}_\ell, \bar{\beta}_h \mid h)$. *Then for any* $\bar{\beta}_h \in [0, 1]$ *and* $\bar{\beta}_\ell \in [0, \frac{q(h|\ell)}{q(\ell|h)} \cdot (1 - \bar{\beta}_h)]$ *(i.e., the range in Lemma 1),* $\Delta u(\bar{\beta}_\ell, \bar{\beta}_h \mid h) \leq \frac{\overline{\Delta PS}}{n-1}$ *only if one of the following two holds: (1)* $\bar{\beta}_h \geq 1 - \bar{b}_h$, *or (2)* $\bar{\beta}_h \leq \bar{b}_h$ *and* $\bar{\beta}_\ell \leq \frac{q(h|\ell)}{q(\ell|h)} \cdot \bar{b}_h$, *where*

$$\bar{b}_h = \frac{4\overline{\Delta PS} \cdot (\Delta h + \Delta \ell + PS(\ell, \mathbf{q}_\ell) - PS(h, \mathbf{q}_\ell))}{(n-1) \cdot (\Delta h + \Delta \ell) \cdot \mathbb{E}_{s \sim \mathbf{q}_h}[PS(s, \mathbf{q}_h) - q(s, \mathbf{q}_\ell)]}.$$

We first consider the case when $\bar{\beta}_\ell = 0$. Note that $\Delta u(0, \bar{\beta}_h \mid h)$ is a quadratic function of $\bar{\beta}_h$ satisfying (1) $\Delta u(0, 1 \mid h) = 0$, (2) $\frac{\partial^2 \Delta u(\bar{\beta}_h, 0 \mid h)}{\partial \bar{\beta}_h^2} = -\frac{2(k-1)}{n-1} \cdot q(h \mid h) \cdot (\Delta h + \Delta \ell) < 0$, and (3) another root other than 1, denoted by $\beta_h''$, satisfies $\beta_h'' \leq 0$. If (3) does not hold, we will have $\Delta u(0, 0 \mid h) < 0$, which is a contradiction.

According to the property of the quadratic function, $\Delta u(\bar{\beta}_\ell, \bar{\beta}_h \mid h)$ is maximized at $\frac{1+\beta_h''}{2} =$

$$\frac{(k-1) \cdot (\Delta \ell + q(h \mid h) \cdot (PS(\ell, \mathbf{q}_\ell) - PS(h, \mathbf{q}_\ell)) + (n-k) \cdot (q(h \mid h) \cdot (-\Delta h) + q(\ell \mid h) \cdot \Delta \ell)}{2(k-1) \cdot (q(h \mid h) \cdot (\Delta h + \Delta \ell))}$$

with value

$$\frac{k-1}{4(n-1)} \cdot q(h \mid h) \cdot (\Delta h + \Delta \ell) \cdot (1 - \beta_h'')^2$$

$$\geq \frac{k-1}{4(n-1)} \cdot q(h \mid h) \cdot (\Delta h + \Delta \ell).$$

We consider three different cases

Firstly, when $PS(\ell, \mathbf{q}_\ell) - PS(h, \mathbf{q}_\ell) \leq 0$. In this case, we show that $u(0, 0 \mid h)$ is faraway from $u(\Sigma^* \mid h)$. Therefore, $u(\Sigma^* \mid h) - u(0, \bar{\beta}_h \mid h) \leq \frac{\overline{\Delta PS}}{n-1}$ only if $\bar{\beta}_h$ is close to 1. Note that in this case,

$$u(0, 0 \mid h, \text{deviator}) = PS(\ell, \mathbf{q}_\ell)$$
$$\leq q(h \mid h) \cdot q(\ell, \mathbf{q}_h) + q(\ell \mid h) \cdot PS(\ell, \mathbf{q}_\ell).$$

Therefore,

$$\Delta u(0, 0 \mid h) = u(\Sigma^* \mid h) - u(0, 0 \mid h)$$
$$\geq q(h \mid h) \cdot \Delta h - q(\ell \mid h) \cdot \Delta \ell$$
$$= \mathbb{E}_{s \sim \mathbf{q}_h}[PS(s, \mathbf{q}_h) - q(s, \mathbf{q}_\ell)]$$
$$> 0.$$

Then, give that $\Delta u(0, \bar{\beta}_h \mid h)$ is convex on $\bar{\beta}_h$, for all $\bar{\beta}_h \in [0, 1]$,

$$\Delta u(0, \bar{\beta}_h \mid h) \geq (1 - \bar{\beta}_h) \cdot \mathbb{E}_{s \sim \mathbf{q}_h}[PS(s, \mathbf{q}_h) - q(s, \mathbf{q}_\ell)].$$

Therefore, $\Delta u(0, \bar{\beta}_h \mid h) > \frac{\overline{\Delta PS}}{n-1}$ for any $0 \leq \bar{\beta}_h < 1 - \frac{\overline{\Delta PS}}{(n-1) \cdot \mathbb{E}_{s \sim \mathbf{q}_h}[PS(s, \mathbf{q}_h) - q(s, \mathbf{q}_\ell)]}$.

Secondly, when $PS(\ell, \mathbf{q}_\ell) - PS(h, \mathbf{q}_\ell) \geq 0$ and $\frac{1+\beta_h''}{2} \leq 0$, we still prove that $u(0, 0 \mid h)$ is faraway from $u(\Sigma^* \mid h)$. Note that when $\frac{1+\beta_h''}{2} \leq 0$, the deviating group size $k$ must satisfy

$$k \leq \frac{\mathbb{E}_{s \sim \mathbf{q}_h}[PS(s, \mathbf{q}_h) - q(s, \mathbf{q}_\ell)]}{q(h \mid h) \cdot (\Delta h + \Delta \ell + PS(\ell, \mathbf{q}_\ell) - PS(h, \mathbf{q}_\ell))} \cdot (n-1) + 1.$$

Then,

$$\Delta u(0, 0 \mid h) = \frac{1}{n-1}\big((k-1)\cdot(u(0, 0 \mid h, \text{truthful}) - u(0, 0 \mid h, \text{deviator}))$$

$$+ (n-1)\cdot(u(\Sigma^* \mid h) - u(0, 0 \mid h, \text{truthful})))$$

$$\geq \frac{1}{q(h \mid h)(\Delta h + \Delta \ell + PS(\ell, \mathbf{q}_\ell) - PS(h, \mathbf{q}_\ell))}$$

$$\cdot (-\mathbb{E}_{s \sim \mathbf{q}_h}[PS(s, \mathbf{q}_h) - q(s, \mathbf{q}_\ell)] \cdot q(h \mid h) \cdot (PS(\ell, \mathbf{q}_\ell) - PS(h, \mathbf{q}_\ell))$$

$$+ q(h \mid h) \cdot (\Delta h + \Delta \ell + PS(\ell, \mathbf{q}_\ell) - PS(h, \mathbf{q}_\ell)) \cdot \mathbb{E}_{s \sim \mathbf{q}_h}[PS(s, \mathbf{q}_h) - q(s, \mathbf{q}_\ell)])$$

$$= \frac{(\Delta h + \Delta \ell) \cdot \mathbb{E}_{s \sim \mathbf{q}_h}[PS(s, \mathbf{q}_h) - q(s, \mathbf{q}_\ell)]}{(\Delta h + \Delta \ell + PS(\ell, \mathbf{q}_\ell) - PS(h, \mathbf{q}_\ell))}$$

$$> 0.$$

Therefore, $\Delta u(0, \bar{\beta}_h \mid h) > \frac{\overline{\Delta PS}}{n-1}$ for any $0 \leq \bar{\beta}_h < 1 - \frac{\overline{\Delta PS} \cdot (\Delta h + \Delta \ell + PS(\ell, \mathbf{q}_\ell) - PS(h, \mathbf{q}_\ell))}{(n-1) \cdot (\Delta h + \Delta \ell) \cdot \mathbb{E}_{s \sim \mathbf{q}_h}[PS(s, \mathbf{q}_h) - q(s, \mathbf{q}_\ell)]}$.

Thirdly, when $PS(\ell, \mathbf{q}_\ell) - PS(h, \mathbf{q}_\ell) \geq 0$ and $\frac{1 + \beta_h''}{2} > 0$, the group size $k$ must satisfy

$$k > \frac{\mathbb{E}_{s \sim \mathbf{q}_h}[PS(s, \mathbf{q}_h) - q(s, \mathbf{q}_\ell)]}{q(h \mid h) \cdot (\Delta h + \Delta \ell + PS(\ell, \mathbf{q}_\ell) - PS(h, \mathbf{q}_\ell))} \cdot (n-1) + 1.$$

Therefore, For any $\beta_h \in [\frac{1 + \beta_h''}{2}, 1]$,

$$\Delta u(0, \bar{\beta}_h \mid h) \geq (1 - \bar{\beta}_h) \cdot \frac{k-1}{4(n-1)} \cdot q(h \mid h) \cdot (\Delta h + \Delta \ell)$$

$$\geq (1 - \bar{\beta}_h) \cdot \frac{\mathbb{E}_{s \sim \mathbf{q}_h}[PS(s, \mathbf{q}_h) - q(s, \mathbf{q}_\ell)] \cdot (\Delta h + \Delta \ell)}{4(\Delta h + \Delta \ell + PS(\ell, \mathbf{q}_\ell) - PS(h, \mathbf{q}_\ell))}$$

Similarly, for any $\beta_h \in [0, \frac{1 + \beta_h''}{2}]$,

$$\Delta u(0, \bar{\beta}_h \mid h) \geq \bar{\beta}_h \cdot \frac{\mathbb{E}_{s \sim \mathbf{q}_h}[PS(s, \mathbf{q}_h) - q(s, \mathbf{q}_\ell)] \cdot (\Delta h + \Delta \ell)}{4(\Delta h + \Delta \ell + PS(\ell, \mathbf{q}_\ell) - PS(h, \mathbf{q}_\ell))}.$$

Therefore, $\Delta u(0, \bar{\beta}_h \mid h) > \frac{\overline{\Delta PS}}{n-1}$ for any $\bar{b}_h < \bar{\beta}_h < 1 - \bar{b}_h$, where

$$\bar{b}_h = \frac{4\overline{\Delta PS} \cdot (\Delta h + \Delta \ell + PS(\ell, \mathbf{q}_\ell) - PS(h, \mathbf{q}_\ell))}{(n-1) \cdot (\Delta h + \Delta \ell) \cdot \mathbb{E}_{s \sim \mathbf{q}_h}[PS(s, \mathbf{q}_h) - q(s, \mathbf{q}_\ell)]}.$$

Then we consider when $\bar{\beta}_\ell = \frac{q(h|h)}{q(\ell|h)} \cdot (1 - \bar{\beta}_h)$. From Lemma 1,

$$\frac{\partial u(\frac{q(h|h)}{q(\ell|h)}(1 - \bar{\beta}_h), \bar{\beta}_h \mid h)}{\partial \bar{\beta}_h} = \mathbb{E}_{s \sim \mathbf{q}_h}[PS(s, \mathbf{q}_h) - PS(s, \mathbf{q}_\ell)] > 0.$$

Therefore, for any $0 \leq \bar{\beta}_h < 1 - \frac{\overline{\Delta PS}}{(n-1) \cdot \mathbb{E}_{s \sim \mathbf{q}_h}[PS(s, \mathbf{q}_h) - q(s, \mathbf{q}_\ell)]}$, $\Delta u(\frac{q(h|h)}{q(\ell|h)} \cdot (1 - \bar{\beta}_h), \bar{\beta}_h \mid h) > \frac{\overline{\Delta PS}}{n-1}$.

Then, by the linearity of $\Delta u(\bar{\beta}_\ell, \bar{\beta}_h \mid h)$ on $\bar{\beta}_\ell$, we know that for any $\bar{b}_h < \bar{\beta}_h < 1 - \bar{b}_h$ and any $0 \leq \bar{\beta}_\ell < \frac{q(h|h)}{q(\ell|h)} \cdot (1 - \bar{\beta}_h)$, $\Delta u(\bar{\beta}_\ell, \bar{\beta}_h \mid h) > \frac{\overline{\Delta PS}}{n-1}$, the threshold comes out that $\bar{b}_h$ is the largest among all the threshold.

Then we consider $0 \leq \bar{\beta}_h \leq \bar{b}_h$. We have

$$\Delta u(0, \bar{\beta}_h \mid h) \geq \bar{\beta}_h \cdot \frac{\mathbb{E}_{s \sim \mathbf{q}_h}[PS(s, \mathbf{q}_h) - q(s, \mathbf{q}_\ell)] \cdot (\Delta h + \Delta \ell)}{4(\Delta h + \Delta \ell + PS(\ell, \mathbf{q}_\ell) - PS(h, \mathbf{q}_\ell))}$$

and

$$\Delta u(\frac{q(h \mid h)}{q(\ell \mid h)} \cdot (1 - \bar{\beta}_h), \bar{\beta}_h \mid h) = (1 - \bar{\beta}_h) \cdot \mathbb{E}_{s \sim \mathbf{q}_h}[PS(s, \mathbf{q}_h) - PS(s, \mathbf{q}_\ell)]$$

$$\geq (1 - \bar{\beta}_h) \cdot \frac{\mathbb{E}_{s \sim \mathbf{q}_h}[PS(s, \mathbf{q}_h) - q(s, \mathbf{q}_\ell)] \cdot (\Delta h + \Delta \ell)}{4(\Delta h + \Delta \ell + PS(\ell, \mathbf{q}_\ell) - PS(h, \mathbf{q}_\ell))}.$$

Therefore, for any $0 \leq \bar{\beta}_\ell \leq \frac{q(h|h)}{q(\ell|h)} \cdot (1 - \bar{\beta}_h)$,

$$\Delta u(\bar{\beta}_\ell, \bar{\beta}_h \mid h) = \left(1 - \frac{\bar{\beta}_\ell \cdot q(\ell \mid h)}{q(h \mid h) \cdot (1 - \bar{\beta}_h)}\right) \cdot \Delta u(0, \bar{\beta}_h \mid h)$$

$$+ \frac{\bar{\beta}_\ell \cdot q(\ell \mid h)}{q(h \mid h) \cdot (1 - \bar{\beta}_h)} \cdot \Delta u\left(\frac{q(h \mid h)}{q(\ell \mid h)} \cdot (1 - \bar{\beta}_h), \bar{\beta}_h \mid h\right)$$

$$\geq \left(\bar{\beta}_h + \bar{\beta}_\ell \cdot \frac{q(\ell \mid h) \cdot (1 - 2\bar{\beta}_h)}{q(h \mid h) \cdot (1 - \bar{\beta}_h)}\right) \cdot \frac{\mathbb{E}_{s \sim \mathbf{q}_h}[PS(s, \mathbf{q}_h) - q(s, \mathbf{q}_\ell)] \cdot (\Delta h + \Delta \ell)}{4(\Delta h + \Delta \ell + PS(\ell, \mathbf{q}_\ell) - PS(h, \mathbf{q}_\ell))}$$

$$\geq \left(\bar{\beta}_h + \frac{q(\ell \mid h)}{q(h \mid h)} \cdot \bar{\beta}_\ell \cdot (1 - 2\bar{\beta}_h)\right) \cdot \frac{\mathbb{E}_{s \sim \mathbf{q}_h}[PS(s, \mathbf{q}_h) - q(s, \mathbf{q}_\ell)] \cdot (\Delta h + \Delta \ell)}{4(\Delta h + \Delta \ell + PS(\ell, \mathbf{q}_\ell) - PS(h, \mathbf{q}_\ell))}.$$

Therefore, for

$$\bar{\beta}_\ell > \frac{q(h \mid h)}{q(\ell \mid h) \cdot (1 - 2\bar{\beta}_h)} \cdot (\bar{b}_h - \bar{\beta}_h),$$

$\Delta u(\bar{\beta}_\ell, \bar{\beta}_h \mid h) > \frac{\overline{\Delta PS}}{n-1}$. Given that $\bar{b}_h < \frac{1}{2}$, the RHS is maximized at $\bar{\beta}_h = 0$. Therefore, for any $0 \leq \bar{\beta}_h \leq \bar{b}_h$ and any $\bar{\beta}_\ell > \frac{q(h|h)}{q(\ell|h)} \cdot \bar{b}_h$, $\Delta u(\bar{\beta}_\ell, \bar{\beta}_h \mid h) > \frac{\overline{\Delta PS}}{n-1}$.

So far we have proved Lemma 5, which ends the second part.

In the third part, for the area close to $(0,0)$ or $(0,1)$, i.e. (1) $\bar{\beta}_h \geq 1 - \bar{b}_h$, and (2) $\bar{\beta}_h \leq \bar{b}_h$ and $\bar{\beta}_\ell \leq \frac{q(h|h)}{q(\ell|h)} \cdot \bar{b}_h$, we show that we can always find a deviator $i$ such that $u(\beta_\ell^i, \beta_h^i \mid h) < u(\Sigma^* \mid h)$. Let

$$\hat{\beta}_h = \frac{1}{k-1} \sum_{j \in D, j \neq i} \beta_h^j = \bar{\beta}_h + \frac{1}{k-1}(\bar{\beta}_h - \beta_h^i)$$

$$\hat{\beta}_\ell = \frac{1}{k-1} \sum_{j \in D, j \neq i} \beta_\ell^j = \bar{\beta}_\ell + \frac{1}{k-1}(\bar{\beta}_\ell - \beta_\ell^i)$$

be the average strategy of all the deviators other than $i$. It is satisfied that $(\hat{\beta}_\ell, \hat{\beta}_h) \in [0,1]^2$.

We pick a deviator $i$ and characterize the $k$ such that $u(\beta_\ell^i, \beta_h^i \mid h) \geq u(\Sigma^* \mid h)$. More precisely,

$$k \geq \frac{u(\Sigma^* \mid h) - u(\beta_\ell^i, \beta_h^i \mid h, \text{truthful})}{(u(\Sigma^* \mid h) - u(\beta_\ell^i, \beta_h^i \mid h, \text{truthful})) - (u(\Sigma^* \mid h) - u(\beta_\ell^i, \beta_h^i \mid h, \text{deviator}))} \cdot (n-1) + 1$$

when the denominator $(u(\Sigma^* \mid h) - u(\beta_\ell^i, \beta_h^i \mid h, \text{truthful})) - (u(\Sigma^* \mid h) - u(\beta_\ell^i, \beta_h^i \mid h, \text{deviator})) > 0$ or $k$ does not exist when the denominator equals to or is less than 0. We will show that this $k > k_B$ in both corner cases.

The numerator of RHS is

$$u(\Sigma^* \mid h) - u(\beta_\ell^i, \beta_h^i \mid h, \text{truthful}) = (1 - \beta_h^i) \cdot \mathbb{E}_{s \sim \mathbf{q}_h}[PS(s, \mathbf{q}_h) - PS(s, \mathbf{q}_\ell)].$$

The denominator is

$$u(\beta_\ell^i, \beta_h^i \mid h, \text{deviator}) - u(\beta_\ell^i, \beta_h^i \mid h, \text{truthful})$$

$$= \beta_h^i \cdot ((q(h \mid h) \cdot (\hat{\beta}_h - 1) + q(\ell \mid h) \cdot \hat{\beta}_\ell) \cdot PS(h, \mathbf{q}_h)$$

$$+ (q(h \mid h) \cdot (1 - \hat{\beta}_h) - q(\ell \mid h) \cdot \hat{\beta}_\ell) \cdot PS(\ell, \mathbf{q}_h))$$

$$= (1 - \beta_h^i) \cdot ((q(h \mid h) \cdot (\hat{\beta}_h - 1) + q(\ell \mid h) \cdot \hat{\beta}_\ell) \cdot PS(h, \mathbf{q}_\ell)$$

$$+ (q(h \mid h) \cdot (1 - \hat{\beta}_h) - q(\ell \mid h) \cdot \hat{\beta}_\ell) \cdot PS(\ell, \mathbf{q}_\ell))$$

$$= (q(h \mid h) \cdot (1 - \hat{\beta}_h) - q(\ell \mid h) \cdot \hat{\beta}_\ell)$$

$$\cdot (\beta_h^i \cdot (PS(\ell, \mathbf{q}_h) - PS(h, \mathbf{q}_h)) + (1 - \beta_h^i) \cdot (PS(\ell, \mathbf{q}_\ell) - PS(h, \mathbf{q}_\ell))).$$

We first consider $\bar{\beta}_h \leq \bar{b}_h$ and $\bar{\beta}_\ell \leq \frac{q(h|h)}{q(\ell|h)} \cdot \bar{b}_h$. In this case, we pick a deviator $i$ such that $\beta_h^i \leq \bar{\beta}_h$.

Firstly, there must be $(q(h \mid h) \cdot (1 - \hat{\beta}_h) - q(\ell \mid h) \cdot \hat{\beta}_\ell) > 0$ for all sufficiently large $n$. This is because $\bar{\beta}_h \leq \bar{b}_h = \Theta(\frac{\overline{\Delta PS}}{n-1})$ and $\bar{\beta}_\ell \leq \frac{q(h|h)}{q(\ell|h)} \cdot \bar{b}_h$. Moreover, $\hat{\beta}_h \leq 2\bar{\beta}_h$ and $\hat{\beta}_\ell \leq 2\bar{\beta}_\ell$ by the property of the average. Therefore, for sufficiently large $n$ such that $\bar{b}_h < \frac{1}{4}$, $(q(h \mid h) \cdot (1 - \hat{\beta}_h) - q(\ell \mid h) \cdot \hat{\beta}_\ell) > 0$.

If $PS(\ell, \mathbf{q}_\ell) - PS(h, \mathbf{q}_\ell) \leq 0$, there must be $(PS(\ell, \mathbf{q}_h) - PS(h, \mathbf{q}_h)) < 0$. In this case $u(\beta_\ell^i, \beta_h^i \mid h, \text{deviator}) - u(\beta_\ell^i, \beta_h^i \mid h, \text{truthful}) < 0$, and for any $k \geq 2$, $i$'s reward will be strictly lower than the truthful reward.

Suppose $PS(\ell, \mathbf{q}_\ell) - PS(h, \mathbf{q}_\ell) > 0$. In this case, the condition for $u(\beta^i_\ell, \beta^i_h \mid h) \geq u(\Sigma^* \mid h)$ is equivalent to $k - 1 \geq$

$$\frac{(1 - \beta^i_h) \cdot \mathbb{E}_{s \sim \mathbf{q}_h}[PS(s, \mathbf{q}_h) - PS(s, \mathbf{q}_\ell)]}{(q(h \mid h) \cdot (1 - \hat{\beta}_h) - q(\ell \mid h) \cdot \hat{\beta}_\ell) \cdot (\beta^i_h \cdot (PS(\ell, \mathbf{q}_h) - PS(h, \mathbf{q}_h)) + (1 - \beta^i_h) \cdot (PS(\ell, \mathbf{q}_\ell) - PS(h, \mathbf{q}_\ell)))} \cdot (n - 1).$$

We show this lower bound is larger than $k_B$. Recall that

$$k_B \leq \frac{\mathbb{E}_{s \sim \mathbf{q}_h}[PS(s, \mathbf{q}_h) - PS(s, \mathbf{q}_\ell)]}{q(h \mid h) \cdot (PS(\ell, \mathbf{q}_\ell) - PS(h, \mathbf{q}_\ell))} \cdot (n - 1) + 1$$

Therefore, it is sufficient to show that

$$\frac{(1 - \beta^i_h) \cdot q(h \mid h) \cdot (PS(\ell, \mathbf{q}_\ell) - PS(h, \mathbf{q}_\ell))}{(q(h \mid h) \cdot (1 - \hat{\beta}_h) - q(\ell \mid h) \cdot \hat{\beta}_\ell) \cdot (\beta^i_h \cdot (PS(\ell, \mathbf{q}_h) - PS(h, \mathbf{q}_h)) + (1 - \beta^i_h) \cdot (PS(\ell, \mathbf{q}_\ell) - PS(h, \mathbf{q}_\ell)))} > 1.$$

Firstly, given that $\beta^i_h \leq \bar{\beta}_h$, there is $\beta^i_h \leq \hat{\beta}_h$. Therefore,

$$(q(h \mid h) \cdot (1 - \hat{\beta}_h) - q(\ell \mid h) \cdot \hat{\beta}_\ell) \leq q(h \mid h) \cdot (1 - \hat{\beta}_h)$$
$$\leq (1 - \beta^i_h) \cdot q(h \mid h).$$

Secondly, by Lemma 3 we have $PS(\ell, \mathbf{q}_h) - PS(h, \mathbf{q}_h) < PS(\ell, \mathbf{q}_\ell) - PS(h, \mathbf{q}_\ell)$. Therefore, $\beta^i_h \cdot (PS(\ell, \mathbf{q}_h) - PS(h, \mathbf{q}_h)) + (1 - \beta^i_h) \cdot (PS(\ell, \mathbf{q}_\ell) - PS(h, \mathbf{q}_\ell)) \leq PS(\ell, \mathbf{q}_\ell) - PS(h, \mathbf{q}_\ell)$.

By combining two parts, we show that every part in the denominator is smaller than the corresponding part in the nominator. Therefore, the threshold for $i$'s reward exceeds the truthful reward $k \geq k_B$, and the equality holds only when $\beta^i_h = \hat{\beta}_h = \hat{\beta}_\ell = 0$. When the equality holds, all other deviators play $(0, 0)$. If $\beta^i_\ell = 0$, the case is covered by Step 2. Otherwise, we consider a different deviator $i$. Then the threshold for the new $i$ will be strictly larger than $k_B$. Therefore, for all $k \leq k_B$, $i$'s reward is strictly lower than the truthful reward.

We then consider the second area $\bar{\beta}_h \geq 1 - \bar{b}_h$. When $PS(h, \mathbf{q}_h) \leq PS(\ell, \mathbf{q}_h)$, we we pick an $i$ such that $\beta^i_h \leq \bar{\beta}_h$ and compare $i$'s reward with the truthful reward. In this case, $0 \leq PS(\ell, \mathbf{q}_h) - PS(h, \mathbf{q}_h) < PS(\ell, \mathbf{q}_\ell) - PS(h, \mathbf{q}_\ell)$. If $q(h \mid h) \cdot (1 - \hat{\beta}_h) - q(\ell \mid h) \cdot \hat{\beta}_\ell \leq 0$, the denominator is non-positive, and for any $k \geq 2$, $i$'s reward cannot exceed the truthful reward. If $q(h \mid h) \cdot (1 - \hat{\beta}_h) - q(\ell \mid h) \cdot \hat{\beta}_\ell > 0$, the denominator is positive. Following similar reasoning for $(\bar{\beta}_\ell, \bar{\beta}_h)$ close to $(0, 0)$ shows that the threshold $k > k_B$.

Otherwise, when $PS(h, \mathbf{q}_h) > PS(\ell, \mathbf{q}_h)$, we compare $i$'s reward with the reward of average strategy $(\bar{\beta}_\ell, \bar{\beta}_h)$. Recall that

$$u(\beta^i_\ell, \beta^i_h \mid h) - u(\bar{\beta}_\ell, \bar{\beta}_h \mid h)$$
$$= \frac{n - k}{n - 1} \cdot f^h(\beta^i_h - \bar{\beta}_h, (0, 1)) + \frac{k - 1}{n - 1} f(\beta^i_h - \bar{\beta}_h, (\bar{\beta}_\ell, \bar{\beta}_h))$$
$$+ \frac{1}{n - 1}(f(\beta^i_h, (\bar{\beta}_\ell, \bar{\beta}_h)) - f(\beta^i_h, (\beta^i_\ell, \beta^i_h)))$$
$$= (\beta^i_h - \bar{\beta}_h) \cdot (\mathbb{E}_{s \sim \mathbf{q}_h}[PS(s, \mathbf{q}_h) - PS(s, \mathbf{q}_\ell)]$$
$$- \frac{k - 1}{n - 1} \cdot (q(h \mid h) \cdot (1 - \bar{\beta}_h) - q(\ell \mid h) \cdot \bar{\beta}_\ell) \cdot (\Delta h + \Delta \ell))$$
$$+ \frac{1}{n - 1} \cdot (q(h \mid h) \cdot (\bar{\beta}_h - \beta^i_h) + q(\ell \mid h) \cdot (\bar{\beta}_\ell - \beta^i_\ell))$$
$$\cdot (\beta^i_h \cdot (PS(h, \mathbf{q}_h) - PS(\ell, \mathbf{q}_h)) + (1 - \beta^i_h) \cdot (PS(h, \mathbf{q}_\ell) - PS(\ell, \mathbf{q}_\ell))).$$

We will assume that $n$ is sufficiently large so that $((1 - \bar{b}_h) \cdot (PS(h, \mathbf{q}_h) - PS(\ell, \mathbf{q}_h)) + \bar{b}_h \cdot (PS(h, \mathbf{q}_\ell) - PS(\ell, \mathbf{q}_\ell)) > 0$. (Recall that $\bar{b}_h = \Theta(\frac{1}{n-1})$).

Note that for the second line, $(q(h \mid h) \cdot (1 - \bar{\beta}_h) - q(\ell \mid h) \cdot \bar{\beta}_\ell) \cdot (\Delta h + \Delta \ell) \leq q(h \mid h) \cdot \bar{b}_h \cdot (\Delta h + \Delta \ell)$. Therefore, for sufficiently large $n$,

$$(\beta^i_h - \bar{\beta}_h) \cdot (\mathbb{E}_{s \sim \mathbf{q}_h}[PS(s, \mathbf{q}_h) - PS(s, \mathbf{q}_\ell)]$$
$$- \frac{k - 1}{n - 1} \cdot (q(h \mid h) \cdot (1 - \bar{\beta}_h) - q(\ell \mid h) \cdot \bar{\beta}_\ell) \cdot (\Delta h + \Delta \ell)) \geq 0.$$

Let

$$m_1 = (\mathbb{E}_{s \sim \mathbf{q}_h}[PS(s, \mathbf{q}_h) - PS(s, \mathbf{q}_\ell)]$$
$$- \frac{k - 1}{n - 1} \cdot (q(h \mid h) \cdot (1 - \bar{\beta}_h) - q(\ell \mid h) \cdot \bar{\beta}_\ell) \cdot (\Delta h + \Delta \ell))$$
$$m_2(\beta^i_h) = (\beta^i_h \cdot (PS(h, \mathbf{q}_h) - PS(\ell, \mathbf{q}_h)) + (1 - \beta^i_h) \cdot (PS(h, \mathbf{q}_\ell) - PS(\ell, \mathbf{q}_\ell))).$$

Then

$$u(\beta_\ell^i, \beta_h^i \mid h) - u(\bar\beta_\ell, \bar\beta_h \mid h)$$

$$= (\beta_h^i - \bar\beta_h) \cdot m_1 + \frac{1}{n-1} \cdot (q(h \mid h) \cdot (\bar\beta_h - \beta_h^i) + q(\ell \mid h) \cdot (\bar\beta_\ell - \beta_\ell^i)) \cdot m_2(\beta_h^i)$$

$$= (\beta_h^i - \bar\beta_h) \cdot (m_1 - \frac{1}{n-1} \cdot q(h \mid h) \cdot m_2(\beta_h^i)) + \frac{1}{n-1} \cdot q(\ell \mid h) \cdot (\bar\beta_\ell - \beta_\ell^i) \cdot m_2(\beta_h^i).$$

Note that $m_1 > 0$ and $\frac{\partial m_2}{\partial \beta_h^i} = \Delta h + \Delta \ell > 0$.

If there exists a deviator $i$ such that $\beta_h^i \le \bar\beta_h$ and $u(\beta_\ell^i, \beta_h^i \mid h) - u(\bar\beta_\ell, \bar\beta_h \mid h) < 0$, we just pick this $i$. Otherwise, if all deviators $j$ with $\beta_h^j \le \bar\beta_h$ has $u(\beta_\ell^i, \beta_h^i \mid h) - u(\bar\beta_\ell, \bar\beta_h \mid h) \ge 0$, then the range of $j$'s strategy $(\beta_\ell^j, \beta_h^j)$ satisfies

$$(\beta_h^j - \bar\beta_h) \cdot (m_1 - \frac{1}{n-1} \cdot q(h \mid h) \cdot m_2(\beta_h^j)) + \frac{1}{n-1} \cdot q(\ell \mid h) \cdot (\bar\beta_\ell - \beta_\ell^j) \cdot m_2(\beta_h^j) \ge 0$$

This directly implies that $\beta_\ell^j < \bar\beta_\ell$ for any $j$ with $\beta_h^j \le \bar\beta_h$.

Now we pick another deviator $i$ such that (1) $\beta_h^i \ge \bar\beta_h$ and (2) for some deviator $j$ with $\beta_h^j < \bar\beta_h$, $(\beta_h^i - \bar\beta_h)(\bar\beta_\ell - \beta_\ell^j) \le (\beta_\ell^i - \bar\beta_\ell)(\bar\beta_h - \beta_h^j)$. If such $i$ does not exist, then for any $i$ with $\beta_h^i > \bar\beta_h$ and any $j$ with $\beta_h^j \le \bar\beta_h$, there is $(\beta_h^i - \bar\beta_h)(\bar\beta_\ell - \beta_\ell^j) > (\beta_\ell^i - \bar\beta_\ell)(\bar\beta_h - \beta_h^j)$. Then,

$$0 = \sum_{i \in D, \beta_h^i > \bar\beta_h} (\beta_h^i - \bar\beta_h) + \sum_{j \in D, \beta_h^j \le \bar\beta_h} (\beta_h^j - \bar\beta_h)$$

$$> \sum_{i \in D, \beta_h^i > \bar\beta_h} \frac{\beta_\ell^i - \bar\beta_\ell}{\sum_{j \in D, \beta_h^j \le \bar\beta_h} (\beta_\ell^j - \bar\beta_\ell)} \cdot \sum_{j \in D, \beta_h^j \le \bar\beta_h} (\beta_h^j - \bar\beta_h) + \sum_{j \in D, \beta_h^j \le \bar\beta_h} (\beta_h^j - \bar\beta_h)$$

$$= \frac{\sum_{j \in D, \beta_h^j \le \bar\beta_h} (\beta_h^j - \bar\beta_h)}{\sum_{j \in D, \beta_h^j \le \bar\beta_h} (\beta_\ell^j - \bar\beta_\ell)} \cdot \left( \sum_{i \in D, \beta_h^i > \bar\beta_h} (\beta_\ell^i - \bar\beta_\ell) + \sum_{j \in D, \beta_h^j \le \bar\beta_h} (\beta_\ell^j - \bar\beta_\ell) \right)$$

$$= 0,$$

which is a contradiction. Therefore, the deviator $i$ we pick always exists.

Now we compare $i$'s reward with the reward of the average strategy. Note that since $\beta_h^i > \bar\beta_h \ge \beta_h^j$, $m_2(\beta_h^i) > m_2(\beta_h^j)$.

$$u(\beta_\ell^i, \beta_h^i \mid h) - u(\bar\beta_\ell, \bar\beta_h \mid h)$$

$$= (\beta_h^i - \bar\beta_h) \cdot (m_1 - \frac{1}{n-1} \cdot q(h \mid h) \cdot m_2(\beta_h^i)) + \frac{1}{n-1} \cdot q(\ell \mid h) \cdot (\bar\beta_\ell - \beta_\ell^i) \cdot m_2(\beta_h^i)$$

$$< (\beta_h^i - \bar\beta_h) \cdot (m_1 - \frac{1}{n-1} \cdot q(h \mid h) \cdot m_2(\beta_h^j)) + \frac{1}{n-1} \cdot q(\ell \mid h) \cdot (\bar\beta_\ell - \beta_\ell^i) \cdot m_2(\beta_h^j)$$

$$\le \frac{\beta_\ell^i - \bar\beta_\ell}{\bar\beta_\ell - \beta_\ell^j} (\bar\beta_h - \beta_h^j) \cdot (m_1 - \frac{1}{n-1} \cdot q(h \mid h) \cdot m_2(\beta_h^j)) + \frac{1}{n-1} \cdot q(\ell \mid h) \cdot (\bar\beta_\ell - \beta_\ell^i) \cdot m_2(\beta_h^j)$$

$$= -\frac{\beta_\ell^i - \bar\beta_\ell}{\bar\beta_\ell - \beta_\ell^j} \left( (\beta_h^j - \bar\beta_h)(m_1 - \frac{1}{n-1} q(h \mid h) \cdot m_2(\beta_h^j)) + \frac{1}{n-1} q(\ell \mid h) \cdot (\bar\beta_\ell - \beta_\ell^j) \cdot m_2(\beta_h^j) \right)$$

$$\le 0.$$

Therefore, we find an $i$ such that $u(\beta_\ell^i, \beta_h^i \mid h) < u(\bar\beta_\ell, \bar\beta_h \mid h) \le u(\Sigma^* \mid h)$.

Consequently, for any $n$ satisfying:

(1) $\bar b_h = \frac{4\overline{\Delta PS} \cdot (\Delta h + \Delta \ell + PS(\ell, \mathbf{q}_\ell) - PS(h, \mathbf{q}_\ell))}{(n-1) \cdot (\Delta h + \Delta \ell) \cdot \mathbb{E}_{s \sim \mathbf{q}_h}[PS(s, \mathbf{q}_h) - q(s, \mathbf{q}_\ell)]} < \frac{1}{4}$,

(2) If $PS(h, \mathbf{q}_h) > PS(\ell, \mathbf{q}_h)$, then $\bar b_h \le \frac{PS(h, \mathbf{q}_h) - PS(\ell, \mathbf{q}_h)}{\Delta h + \Delta \ell}$,

(3) $\bar b_h \le \frac{\mathbb{E}_{s \sim \mathbf{q}_h}[PS(s, \mathbf{q}_h) - q(s, \mathbf{q}_\ell)]}{q(h|h) \cdot (\Delta h + \Delta \ell)}$,

for any deviation with no more than $k_B$ deviators and the average strategy in the area of Lemma 1, there exists a deviator $i$ with private signal $h$ whose reward is strictly worse than the truthful reward. Therefore, such deviation cannot succeed.

Similarly, for the $\ell$ side, for any $n$ such that

(1) $\bar b_l = \frac{4\overline{\Delta PS} \cdot (\Delta h + \Delta \ell + PS(h, \mathbf{q}_h) - PS(\ell, \mathbf{q}_h))}{(n-1) \cdot (\Delta h + \Delta \ell) \cdot \mathbb{E}_{s \sim \mathbf{q}_\ell}[PS(s, \mathbf{q}_\ell) - q(s, \mathbf{q}_h)]} < \frac{1}{4}$,

(2) If $PS(\ell, \mathbf{q}_\ell) > PS(h, \mathbf{q}_\ell)$, then $\bar b_h \le \frac{PS(\ell, \mathbf{q}_\ell) - PS(h, \mathbf{q}_\ell)}{\Delta h + \Delta \ell}$,

(3) $\bar{b}_h \leq \frac{\mathbb{E}_{s \sim \mathbf{q}_\ell}[PS(s,\mathbf{q}_\ell) - q(s,\mathbf{q}_h)]}{q(\ell|\ell) \cdot (\Delta h + \Delta \ell)}$,

for any deviation with no more than $k_B$ deviators and the average strategy in the area of Lemma 2, there exists a deviator $i$ with private signal $\ell$ whose reward is strictly worse than the truthful reward. Therefore, such deviation cannot succeed.

Therefore, truthful reporting is an Bayesian $k_B$-strong equilibrium.

## E  Truthful Reporting is not a coalitional interim equilibrium

In this section, we introduce the coalitional interim equilibrium in [26].

**Definition 5.** Given the set of all admissible deviating groups $\mathcal{D}$, a strategy profile $\Sigma$ is an interim $\mathcal{D}$ equilibrium if there does not exist a group of agent $D \in \mathcal{D}$, a set of types $s_D = (s_i)_{i \in D}$, and a different strategy profile $\Sigma' = (\sigma_i')$ such that

(1) for all agent $i \notin D$, $\sigma_i' = \sigma_i$;

(2) for all $i \in D$, $u_i(\Sigma' \mid s_D) > u_i(\Sigma \mid s_D)$,

where $u_i(\Sigma \mid s_D)$ is $i$'s expected utility conditioned on he/she knows the types of all the deviators in $D$.

When $\mathcal{D} = \{\{i\} \mid i \in [n]\}$ contains only singletons, interim $\mathcal{D}$ equilibrium is exactly the Bayesian Nash equilibrium. On the other hand, we show that for any $\mathcal{D}$ containing a group of at least two agents, truthful reporting fails to be an interim $\mathcal{D}$ equilibrium.

**Proposition 4.** In the peer prediction mechanism, assume $PS(h, \mathbf{q}_h) > PS(\ell, \mathbf{q}_h)$ and $PS(\ell, \mathbf{q}_\ell) > PS(h, \mathbf{q}_\ell)$. Then for any constant $d \geq 2$ any $\mathcal{D}$ such that there exists a $D \in \mathcal{D}$ with $|D| = d$, and for all sufficiently large $n$, truthful reporting is NOT an interim $\mathcal{D}$ equilibrium.

Proof.  Let $D \in \mathcal{D}$ such that $|D| = d$ be a deviate group. Suppose there are $d_1 > 0$ agents with signal $h$ and $d_2 > 0$ agents with signal $\ell$. $d_1 + d_2 = |D|$.

Now consider the expected utility when every agent reports truthfully. For agent $i$ with signal $h$, $i$'s reward from other deviators is $\frac{d_1-1}{n-1} \cdot PS(h, \mathbf{q}_h) + \frac{d_2}{n-1} \cdot PS(\ell, \mathbf{q}_h)$. And $i$'s reward from truthful reporter is $\frac{n-|D|}{n-1} \cdot (q(h \mid s_D) \cdot PS(h, \mathbf{q}_h) + q(\ell \mid s_D) \cdot PS(\ell, \mathbf{q}_h))$. Similarly, with agent $i$ with signal $\ell$, $i$'s reward from other deviators is $\frac{d_1}{n-1} \cdot PS(h, \mathbf{q}_\ell) + \frac{d_2-1}{n-1} \cdot PS(\ell, \mathbf{q}_\ell)$. And $i$'s reward from truthful reporter is $\frac{n-|D|}{n-1} \cdot (q(h \mid s_D) \cdot PS(h, \mathbf{q}_\ell) + q(\ell \mid s_D) \cdot PS(\ell, \mathbf{q}_\ell))$.

Now we consider the deviating strategy. If $q(h \mid s_D) \cdot PS(h, \mathbf{q}_h) + q(\ell \mid s_D) \cdot PS(\ell, \mathbf{q}_h) > q(h \mid s_D) \cdot PS(h, \mathbf{q}_\ell) + q(\ell \mid s_D) \cdot PS(\ell, \mathbf{q}_\ell)$, then all the deviators report $h$. If $q(h \mid s_D) \cdot PS(h, \mathbf{q}_h) + q(\ell \mid s_D) \cdot PS(\ell, \mathbf{q}_h) < q(h \mid s_D) \cdot PS(h, \mathbf{q}_\ell) + q(\ell \mid s_D) \cdot PS(\ell, \mathbf{q}_\ell)$, then all the deviators report $\ell$. If $q(h \mid s_D) \cdot PS(h, \mathbf{q}_h) + q(\ell \mid s_D) \cdot PS(\ell, \mathbf{q}_h) = q(h \mid s_D) \cdot PS(h, \mathbf{q}_\ell) + q(\ell \mid s_D) \cdot PS(\ell, \mathbf{q}_\ell)$, all the deviators report $h$ if $PS(h, \mathbf{q}_h) \geq PS(\ell, \mathbf{q}_\ell)$ and report $\ell$ otherwise. We show that in this case, the deviation succeeds.

**Case 1.** Suppose $q(h \mid s_D) \cdot PS(h, \mathbf{q}_h) + q(\ell \mid s_D) \cdot PS(\ell, \mathbf{q}_h) > q(h \mid s_D) \cdot PS(h, \mathbf{q}_\ell) + q(\ell \mid s_D) \cdot PS(\ell, \mathbf{q}_\ell)$. We consider the changes on the expected utility after the deviators switch from truthful reporting to the deviating strategy. Then for agents with signal $h$, the expected utility from the truthful reporters is unchanged, and the expected utility from other deviators becomes $\frac{d-1}{n-1} PS(h, \mathbf{q}_h)$, which has been strictly increased. For agents with signal $\ell$, the expected utility from the truthful reporters strictly increases by a constant factor, while the changes in expected utility from other deviators is $\Theta(\frac{d}{n}) = \frac{1}{n}$). Therefore, the sufficiently large $n$, the expected utility for agents with signal $\ell$ also strictly increases.

**Case 2** follows similar reasoning to Case 1.

**Case 3**. Suppose $q(h \mid s_D) \cdot PS(h, \mathbf{q}_h) + q(\ell \mid s_D) \cdot PS(\ell, \mathbf{q}_h) = q(h \mid s_D) \cdot PS(h, \mathbf{q}_\ell) + q(\ell \mid s_D) \cdot PS(\ell, \mathbf{q}_\ell)$ In this case, for both type of agents, the expected utility from truthful reporters is unchanged, and the expected utility from other deviators strictly increases.

Therefore, we show that in all cases, there exists a group of agents in $\mathcal{D}$ wish to deviate. Therefore, truthful reporting is not an interim $\mathcal{D}$ equilibrium. □

Received 20 February 2007; revised 12 March 2009; accepted 5 June 2009

