# OpenReview forum: "Strong Equilibria in Bayesian Games with Bounded Group Size"
_ACM.org/TheWebConf/2025/Conference — WWW 2025 Poster_

### Official Review · Reviewer_FV5r · 2024-11-08

**Novelty:** 6
**Technical Quality:** 6

**Review:**

This paper proposes two solution concepts for Bayesian games with incomplete information:
ex-ante Bayesian k-strong equilibrium and Bayesian k-strong equilibrium. The goal is to capture the possible collusion among game participants. Truthful reporting is ex-ante Bayesian k-strong equilibrium if no group of k agents can collude to gain higher utility before learning their types.  Truthful reporting is Bayesian k-strong equilibrium if no group of k agents can collude to gain higher utility after learning their types.

The peer prediction mechanism is the targeted application of the solution concepts. The main technical results characterise the lower bounds for the required number of colluding agents to gain higher utility in the ex-ante and interim sense. Although section 4 mentioned two more possible applications of the solution concepts, no explicit derivation of any bound is given.

**Questions:**

Is there a proper scoring rule for peer prediction mechanism that leads to (ex-ante) Bayesian n-strong equilibrium?

**Reviewer Confidence:**

3: The reviewer is confident but not certain that the evaluation is correct

**Scope:**

4: The work is relevant to the Web and to the track, and is of broad interest to the community

---

### Official Review · Reviewer_oVPC · 2024-11-19

**Novelty:** 5
**Technical Quality:** 3

**Review:**

The author of this article focuses on the group strategic behaviors in Bayesian games, taking into account the universality of previous theories and considering the case of group strategic behaviors with bounded sizes, proposing a threshold for group size and delving into small-scale joint Bayesian games.\
Pros:
1. This article considers for the first time the strategic behavior of bounded group sizes, providing a new perspective and tool for analyzing group strategic behavior in Bayesian games, and filling the gap in previous research on strong equilibrium of unbounded group sizes.
2. The paper fully demonstrates the practical application value of the proposed equilibrium concept through the specific scenario of peer prediction mechanism. By accurately calculating the population size threshold for the equilibrium of true reports, a clear standard is provided for evaluating the robustness of peer prediction mechanisms to collusion. In addition, the paper also explores other potential application scenarios such as voting and Blotto games, further demonstrating the broad applicability of the proposed equilibrium concept.
3. The paper adopts rigorous mathematical methods and game theory tools to define and prove the new concept of equilibrium. From building models, setting hypotheses to conducting theoretical deductions and proofs, the paper has formed a complete theoretical system.

Cons:
1. The paper adopted some simplifying assumptions when constructing the model, such as symmetric common priors, finite signal sets, and action sets. Although these assumptions facilitate theoretical analysis, they may limit the applicability of the model to complex real-world situations. In practical scenarios, the common prior may not be asymmetric, and the signal set and action set may also be more diverse and complex.
2. Although the paper discusses application scenarios such as peer prediction mechanisms, voting, and Blotto games, these discussions are relatively preliminary and do not delve into the application of these equilibrium concepts in other more complex or different types of Bayesian games. Future research can further expand application scenarios and validate and enhance the universality and practicality of these concepts through empirical research.
3. The paper mainly focuses on static Bayesian games and does not involve situations in dynamic games. In dynamic games, the actions and strategies of agents will be adjusted based on the actions of other agents, which may introduce more complex collusion and equilibrium problems. Future research can explore the concept and properties of strong equilibrium in dynamic games.
4. The paper points out that in some cases, strong equilibrium may not exist. Although the concept of strong equilibrium with bounded group size has been proposed, the paper does not provide a general method or sufficient discussion on how to construct such equilibrium or prove its existence. This limits the application of these concepts in practical game design, and future research needs to make breakthroughs in this area.
5. The paper is mainly based on theoretical analysis, lacking experimental verification or simulation research to support its conclusions. Through actual experiments or simulations, the performance of these equilibrium concepts in real scenarios can be more intuitively demonstrated, and potential problems and improvement directions can be identified. Future research can strengthen experimental verification and simulation studies to enhance the reliability and practicality of conclusions.

**Questions:**

1. What are the differences between the ex-ante Bayesian 𝑘-strong equilibrium and the Bayesian 𝑘-strong equilibrium proposed in the article, and are they the differences between static games and dynamic games? The article does not clearly explain the differences between the two in the decision-making process.
2. There are multiple examples in the article, some of which are scene distances and some are theoretical explanations. The theoretical explanations overlap with the Theorem section in the article. Can the theoretical part be integrated into the Theorem section?
3. What effects can the new theory have, and how can it optimize compared to previous theories? The article does not clearly state this aspect. Please present a more vivid demonstration of the superiority of the new theory through charts and other forms.
4. The main body of the article is purely theoretical proof, without experimental verification of the actual effect of the theory, and its relevance to ACM is not high. It is recommended to publish it in a journal in the field of mathematics.

**Reviewer Confidence:**

3: The reviewer is confident but not certain that the evaluation is correct

**Scope:**

2: The connection to the Web is incidental, e.g., use of Web data or API

---

### Official Review · Reviewer_QLAw · 2024-11-30

**Novelty:** 5
**Technical Quality:** 5

**Review:**

The paper studies group manipulations in Bayesian games, and proposes two notions of group-sized Nash Equilibrium. The peer prediction mechanism is studied as a special case. The authors provide the largest group size that is free from group manipulation. Other models such as voting are also studied.

Pros:
1. The model and concepts are generalizable.
2. The peer prediction mechanism result in Section 3 is interesting: it provides the tight upper bounds to be free from group manipulation.

Cons:
1. What motivates the definition of proper scoring rule? It is nice that example 2 gives some examples of proper scoring rule. But I'm confused how to understand the definition. Why is it a bad idea if a scoring rule is not "proper"? It would benefit if authors can discuss the definition.
2. Typo. Line 404: Brier -> do you mean proper?

**Questions:**

In Theorem 1 and theorem 2, the tight bound depends on the scoring rule. Does it motivate us to choose a scoring rule such that it is not only "proper", but also makes the bound as large as possible?

**Reviewer Confidence:**

2: The reviewer is willing to defend the evaluation, but it is likely that the reviewer did not understand parts of the paper

**Scope:**

4: The work is relevant to the Web and to the track, and is of broad interest to the community

---

### Official Review · Reviewer_2Fg3 · 2024-12-01

**Novelty:** 4
**Technical Quality:** 4

**Review:**

The authors propose a relaxation of strong Bayesian Nash equilibria. The latter allows for multi-lateral deviation by groups of any size. The authors propose k-strong Bayesian Nash equilibria (k-SBNE) as well as an ex-ante strengthening of that concept (k-ESBNE) that precludes multi-lateral deviation among groups of size k. They analyze this concept for peer prediction games and provide thresholds on the value of k above which truthful reporting is not k-SBNE or k-ESBNE. They argue similar results can be obtained for other games such as private Blotto games and voting with partially informed voters. Generally, SBNE can be too strong and not exists while k-ESBNE or k-SBNE do exist and may be more practically relevant given the difficulty of coordinating multi-lateral deviation in large groups.

The argument about k-ESBNE and k-SBNE reflecting the risk attitude of agents towards deviations is concerning. The risk attitude of agents will be reflected in their utility and it is not clear why deviation from equilibrium entails risk. it is clear that k-ESBNE is a stronger concept than k-SBNE, and the natural thing to do would be to give examples of games where k-ESBNE does not exist but k-SBNE does. The authors' analysis of peer prediction provides an avenue by setting game parameters such that k-ESBNE does not exist for any k but k-SBNE does. Another natural approach is to provide examples (based on the timing of the agents and nature's moves) where agents know or do not know their types when coordinating their deviation, making either of the proposed notions suitable based on the information available to agents when forming multi-lateral deviation groups.

Overall, I find the analysis of k-SBNE and k-ESBNE for peer prediction games encouraging but it seems this work can benefit from deeper analysis of a wider class of games or more cases to give a more convincing picture aligned with the claims and introduction that argue for generic relevance of the proposed equilibrium concepts.

**Questions:**

1 - Can you give examples of games where k-ESBNE does not exist but k-SBNE does?

2 - Can you provide examples (based on the timing of the agents and nature's moves) where agents know or do not know their types when coordinating their deviation, making either of the proposed notions suitable based on the information available to agents when forming multi-lateral deviation groups?

**Reviewer Confidence:**

3: The reviewer is confident but not certain that the evaluation is correct

**Scope:**

3: The work is somewhat relevant to the Web and to the track, and is of narrow interest to a sub-community